# A tale of two tails: Preferred and anti-preferred natural stimuli in visual cortex

**Rabia Gondur**[1]    **Patricia L. Stan**[2]    **Matthew A. Smith** [2]    **Benjamin R. Cowley**[1]

[1]Cold Spring Harbor Laboratory, Cold Spring Harbor, NY, USA
[2]Carnegie Mellon University, Pittsburgh, PA, USA
{gondur,cowley}@cshl.edu

## Abstract

An ongoing quest in neuroscience is to find the preferred stimulus of a sensory neuron. This search lays the foundation for understanding how selectivity emerges in the primate visual stream—from simple edge-detecting neurons to highly-selective face neurons—as well as for the architectures and activation functions of deep neural networks. The prevailing notion is that a visual neuron primarily responds to a single preferred visual feature, like an oriented edge or the shape of an object, resulting in a "one-tailed" distribution of responses to natural images. However, surprisingly, we instead find "two-tailed" response distributions of primate visual cortical neurons, suggesting that these neurons have both preferred *and* anti-preferred stimuli. We experimentally validated anti-preferred stimuli by recording responses from macaque V4 to model-optimized stimuli. We find that these anti-preferred stimuli are important for describing a neuron's tuning, as both preferred and anti-preferred images are needed to predict a neuron's responses to natural images. Moreover, in a psychophysics task, humans rely on anti-preferred images to interpret and predict V4 stimulus tuning; this was not the case for internal units from a deep neural network. Interestingly, we find no discernible differences in image statistics between preferred and anti-preferred images. This suggests that by encoding anti-preferred features, a V4 population seemingly doubles its capacity for feature selectivity, allowing for a more flexible downstream readout. Overall, we establish anti-preferred stimuli as an important encoding property of V4 neurons. Our work embarks on a new quest in neuroscience to search for anti-preferred stimuli along the visual stream and offers a new perspective on how feature selectivity arises in the visual cortex and deep neural networks.

## 1 Introduction

Since the first recordings of action potentials from sensory neurons (Hartline, 1938; Lettvin et al., 1940), neuroscientists have searched for the stimulus features that a neuron prefers. Hubel and Wiesel famously identified the stimulus preferences of early visual cortical (V1) neurons as oriented edges (Hubel and Wiesel, 1962). In higher-order visual cortex are neurons with remarkable selectivity (Perrett et al., 1982), such as "Jennifer Aniston" neurons that only respond to images of the celebrity, regardless of profile or hairstyle (Quiroga et al., 2005). This has spurred on new modeling approaches to identify a visual neuron's preferred stimulus—the stimulus that maximizes a neuron's response (Földiák, 2001; Yamane et al., 2008; Okazawa et al., 2015; Cowley et al., 2017a; Cadieu et al., 2007; Abbasi-Asl et al., 2018; Pospisil et al., 2018; Ponce et al., 2019; Bashivan et al., 2019a; Walker et al., 2019; Gu et al., 2022; Cowley et al., 2026; Willeke et al., 2023; Pierzchlewicz et al., 2024; Wang and Ponce, 2024). The classic linear-nonlinear (LN) model, used to describe retinal ganglion cells and simple V1 neurons (Hubel and Wiesel, 1962; Heeger, 1992; Rust et al., 2005; Pillow et al., 2008), exemplifies the concept of a preferred stimulus: The LN model filters visual input to detect a single stimulus pattern (e.g., a localized, oriented edge). The presence of the pattern causes the activity to surpass a ReLU-like threshold, while all other stimulus patterns fail to reach this threshold, silencing the output. This results in a "one-tailed" response distribution (Fig. 1**a**, top row). The deeper units in a task-driven DNN—made up of cascading layers of LN models—achieve the sparse selectivity found in higher-order visual

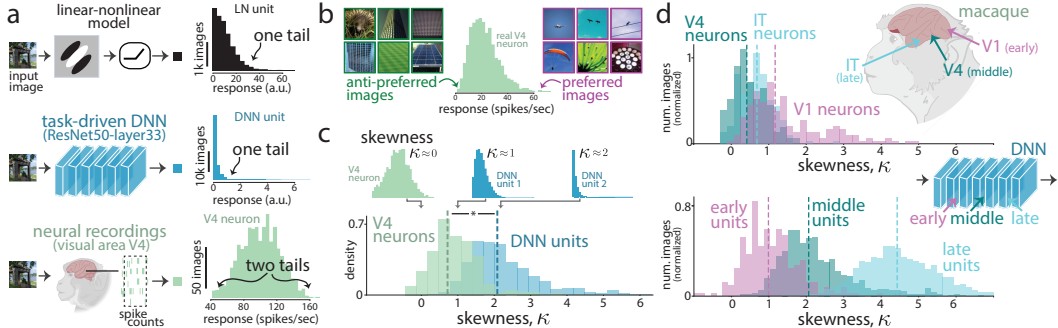

**Figure 1: V4 neurons have two-tailed response distributions. a**. Response distributions for a linear-nonlinear filter (*top*), DNN unit (*middle*), and a real neuron from visual area V4 (*bottom*). **b**. Anti-preferred and preferred images of an example V4 neuron. **c**. Skewness $\kappa$ of response distributions for V4 neurons and DNN units. **d**. Skewness $\kappa$ of response distributions for different visual areas in macaque *(top)*, and DNN layers *(bottom)*. Dashed lines: medians.

cortex. Indeed, the response distributions of post-ReLU DNN units in deeper layers typically have one extreme tail (Fig. 1**a**, middle row) with a few select stimuli evoking large responses. However, when we recorded from real neurons in macaque V4, a higher-order visual area known for encoding texture, shape, color, etc. (Pasupathy et al., 2020), we expected to see similar one-tailed response distributions to natural images. Instead, we unexpectedly found response distributions with two distinct tails (Fig. 1**a**, bottom row, example real V4 neuron). This suggests that, unlike LN models and most DNN units, V4 neurons have preferred (response-maximizing) *and* anti-preferred (response-minimizing) stimuli.

The existence of anti-preferred images for higher-order visual cortical neurons is not obvious. The anti-preferred stimuli have largely been investigated as part of a neuron's tuning for a single stimulus parameter (e.g., orientation: a vertical edge may drive a V1 neuron's response while a horizontal edge suppresses it) (Hubel and Wiesel, 1962; Cavanaugh et al., 2002). However, little is known about the anti-preferred stimuli of V4 neurons when considering the vast space of natural images varying over many stimulus parameters (Efird et al., 2024). Our prior expectations that the anti-preferred images are mostly featureless and low contrast—a blank, gray screen—were wrong; we find that some anti-preferred visual features are as vivid as those for the preferred images (Fig. 1**b**). This motivated us to systematically investigate the existence of anti-preferred images and their roles in how the visual cortex encodes natural images with the following experiments and analyses:

1. We first set out to confirm the existence of anti-preferred images by analyzing response distributions of visual cortical neurons from V1, V4, and IT as well as performing our own electrophysiological experiments to validate that anti-preferred images suppress V4 responses.

2. We find that two-tailed pre-ReLU DNN units are worse predictors of real V4 neurons than one-tailed post-ReLU DNN units; we explain this discrepancy by proposing a new mapping between DNN units and V4 neurons that takes advantage of anti-preferred features to outperform commonly-used linear mappings.

3. We further find that anti-preferred images are integral to a V4 neuron's tuning: Training an encoding model solely on responses to preferred images performs worse than training on responses to both preferred and anti-preferred images. In a similar vein, humans performing a psychophysics task also rely on anti-preferred images to infer a neuron's tuning.

4. How do anti-preferred features contribute to encoding natural images by a V4 population? We find that knowing a neuron's anti-preferred feature provides little information about the neuron's preferred feature but that both have similar image statistics. This suggests that anti-preferred images effectively double the population's capacity for feature selectivity.

5. To encourage further experiments investigating anti-preferred images in the visual cortex, we release a tool called *ImageBeagle* that efficiently "hunts" through millions of natural images. We tailored ImageBeagle for closed-loop, real-time experiments.

Our results change our prior conceptions about stimulus encoding in primate visual cortex: Conceptually, responses are not simply the output of a ReLU with a strong threshold but rather the sum of a baseline offset and a stimulus drive that may enhance or suppress the baseline response, resulting in a two-tailed response distribution. That preferred and anti-preferred features are diverse and distributed across neurons allows the neural population to seemingly double its feature selectivity, providing a rich yet compact basis for readout by downstream IT neurons to carry out object recognition and other visual tasks. Our work speaks to neuroscientists studying how feature selectivity arises in the visual cortex as well as to neuroAI researchers aiming to build biologically-realistic AI models with internal representations that align with those of the visual cortex (Richards et al., 2019; Doerig et al., 2023).

## 2 HIGHER-ORDER VISUAL CORTICAL NEURONS IN AREA V4 HAVE TWO-TAILED RESPONSE DISTRIBUTIONS.

To quantify the degree to which V4 responses to natural images have distributions with two tails (a hallmark of a neuron having preferred and anti-preferred images), we computed the skewness $\kappa$ of response distributions. A distribution with $\kappa$ close to 0 indicates two tails (Fig. 1**c**, top left panel) while $\kappa$ close to 2 indicates a one-tailed distribution (Fig. 1**c**, top right panel). As expected, the skewness for post-ReLU DNN units in a middle layer of a task-driven DNN (ResNet50) (He et al., 2016), known to be predictive of V4 responses (Cowley et al., 2026; Yamins and DiCarlo, 2016; Schrimpf et al., 2018; Bashivan et al., 2019b; Zhuang et al., 2021; Cadena et al., 2024), was close to 2 (Fig. 1**c**, 'DNN units', median $\kappa = 2.06$), indicating that most units in a task-driven DNN have one-tailed response distributions and are selective for one type of visual feature; we confirmed this was true of units from other task-driven DNNs (see Appendix B.2). On the other hand, the response distributions of V4 neurons were better described as two-tailed (Fig. 1**c**, 'V4 neurons', median $\kappa = 0.87$, $p < 0.002$, $n = 219$), suggesting that V4 neurons have both preferred and anti-preferred images (See A.1 in Methods for a description of V4 data collection; V4 responses were repeat-averaged spike counts). Here, we avoid the trivial effects of adaptation (Kohn, 2007)—in which presenting the same image continuously for long periods of time would lead to response suppression—by taking spike counts in a 100 ms bin after the stimulus onset of a natural image preceded by a gray, blank screen. V4 neurons appear to dynamically increase their baseline firing rate to encode a newly presented image (Pasupathy and Connor, 1999; Maunsell, 2015), allowing images to both excite and suppress their response from baseline (investigated in the next section). This goes against the conventional notion that a visual neuron responds selectively to certain stimuli by discarding most other stimuli that fail to drive the neuron past its spiking threshold. In other words, V4 responses do not appear to be the output of ReLU-like activation functions.

These findings motivated us to further investigate whether neurons from other areas of visual cortex also exhibit two-tailed response distributions. We used publicly-available datasets to compute skewness for V1 neurons (Cadena et al., 2019) as well as for V4 and IT neurons (Majaj et al., 2015). We found that skewness values from this other V4 dataset matched our own data (Fig. 1**d**, 'V4 neurons', median $\kappa = 0.41$). In addition, we found that neurons from V1 and IT also exhibit two-tailed selectivity (Fig. 1**d**, 'V1 neurons', median $\kappa = 1.17$ and 'IT neurons', median $\kappa = 0.69$). In contrast, the activations from increasingly-deeper layers of ResNet50 exhibited much larger skewness values. DNN units in an early layer had the lowest skewness (Fig. 1**d**, 'early units', median $\kappa = 0.99$) on par with that observed for V1 neurons. A late layer had the highest skewness value (Fig. 1**d**, 'late units', median $\kappa = 4.43$), revealing a trend of increasing skewness (or one-tailedness) deeper into the network. Taken together, our results indicate a mismatch between biological and artificial visual systems: Neurons along the visual cortical hierarchy tend to have two-tailed response distributions, whereas DNN units in the deepest layers are most likely to have one-tailed response distributions. In other words, most real neurons encode anti-preferred images, but DNN units (post-ReLU) often encode only preferred features, especially in deeper layers.

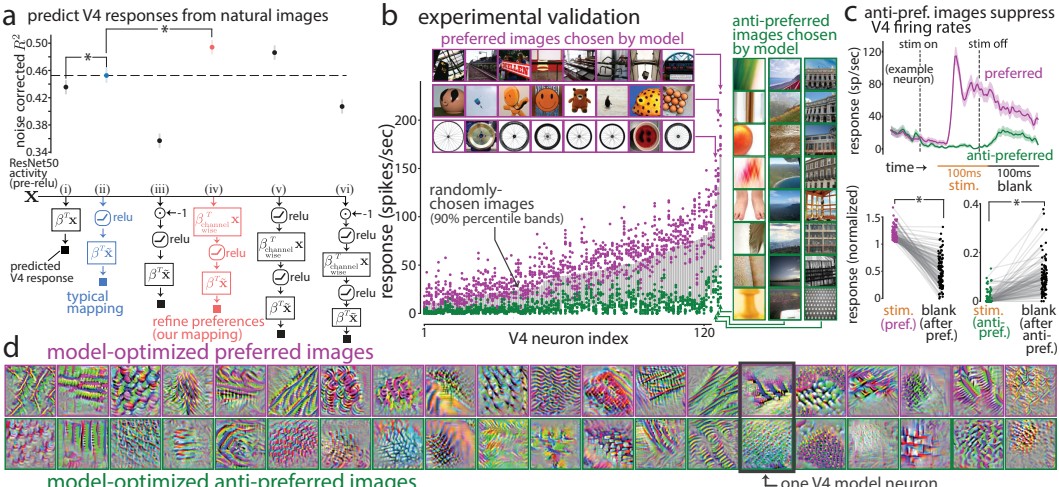

**Figure 2: Experimental evidence that V4 neurons have anti-preferred images. a**. Predicting V4 responses to randomly-chosen images from a linear mapping of ResNet-50 features. Each dot denotes the median, and error bars denote 1 s.e.m. **b**. Experimental validation of preferred and anti-preferred images as predicted by V4 model neurons. Each dot is the repeat-averaged V4 response to one image; gray bands denote 90% percentiles of responses to randomly-chosen natural images. Insets: Model-chosen images for the 3 V4 neurons with largest baseline responses. **c**. Top: Repeat-averaged temporal response of an example V4 neuron (PSTH). Bottom: Normalized response to preferred, anti-preferred, and following blank images; each dot denotes the average response of the top 10 images for one V4 neuron (*bottom*). **d**. Preferred and anti-preferred images synthesized via gradient techniques, one for each V4 model neuron. Asterisks in **a** and **c** denote $p<0.001$, permutation test, $n>50$.

## 3 EXPERIMENTAL EVIDENCE FOR ANTI-PREFERRED IMAGES

A common practice to predict V4 responses to natural images is to linearly map task-driven DNN features to the V4 responses (Yamins and DiCarlo, 2016; Schrimpf et al., 2018). Interestingly, previous studies using this approach almost always use DNN units from a ReLU layer (i.e., post-ReLU) versus a pre-ReLU layer (i.e., the output of convolutional filters not yet passed through ReLUs) (Bashivan et al., 2019a; Cowley et al., 2026; Cadena et al., 2024). Indeed, we find that using a linear mapping to predict V4 responses with post-ReLU activity outperforms the pre-ReLU activity directly preceding the ReLU activation functions (Fig. 2**a**, *i* vs *ii*). On one hand, this was unexpected—presumably, the activity of pre-ReLU DNN units better matches the two-tailedness of real V4 responses; on the other hand, we would expect this performance increase if the pre-ReLU DNN units did not match the real V4 neurons in both preferred and anti-preferred features. In other words, the post-ReLU activity forms a bank of one-tailed preferred features that can be linearly combined to form two-tailed preferred and anti-preferred features. Not every bank of one-tailed preferred features performs well: Flipping the sign before passing the activity through a ReLU— only giving access to the anti-preferred features of the pre-ReLU units—is the poorest predictor (Fig. 2**a**, *iii*), suggesting that the anti-preferred features in task-driven DNNs do not match feature preferences in V4. We conclude that post-ReLU activity better predicts V4 responses not because the activity is more biologically realistic but rather because one can form many possible pairs of preferred and anti-preferred features by mixing and matching post-ReLU "one-tailed" DNN units.

That the pre-ReLU activity performs worse despite having strictly more information (two tails) than the post-ReLU activity (one tail) motivates a new mapping that is agnostic to possible mismatches in preferred and anti-preferred features. We propose to linearly combine filter channels (i.e., a convolution with kernel shape $1 \times 1$ and output channels equal to the number of input channels) and then pass the resulting activity through ReLUs and a final linear mapping. This simple approach significantly improved prediction for pre-ReLU activity (Fig. 2**a**, *iv* vs. *i*) as well as post-ReLU activity (Fig. 2**a**, *iv* vs. *ii*). The extra layer of ReLUs in our proposed mapping also improved post-ReLU activity performance (Fig. 2**a**, *v* vs. *ii*), leading to an expected similar prediction

performance between pre-ReLU and post-ReLU activity (Fig. 2**a**, *iv* vs. *v*, see Methods A.2 for results from other task-driven DNNs). Thus, this new mapping accounts for a possible mismatch in anti-preferred features. If the pre-ReLU DNN units had matching anti-preferred features with the real V4 neurons, our mapping will always outperform using only post-ReLU DNN units by accessing both the preferred and anti-preferred features of the pre-ReLU DNN units (see Supp. Fig. 1). Overall, these findings highlight an important representational gap between task-driven DNNs and biological networks, and offer a simple mapping to overcome this gap.

The skewness of response distributions and the improved prediction by mixing preferences before the ReLUs both hint at the existence of anti-preferred images; we next seek experimental evidence. We build upon recent work that identified highly-predictive DNN models of V4 neurons by training on responses to many natural images (Cowley et al., 2026); these data-driven models predicted the preferred images of real V4 neurons in validation experiments by presenting the model-chosen preferred images in a following recording session (a causal test). We wondered whether the same framework could be used to predict neurons' anti-preferred images. To test this, we recorded V4 responses while the awake, fixating animal (macaque monkey) watched flashes of many images over multiple recording sessions (see Methods A.1), and used the image-response pairs to train a set of data-driven DNN models (which we refer to as V4 model neurons). We identified each V4 model neuron's preferred and anti-preferred images by passing as input 500,000 natural images and keeping the 10 images that either maximized or minimized the model's output response (example chosen images in Fig. 2**b**). We then experimentally validated these predictions by recording V4 responses to these model-chosen preferred and anti-preferred images, along with hundreds of randomly-chosen natural images. After matching V4 neurons to their corresponding model units, we found that the predicted preferred images resulted in responses above the 90% density interval of responses to randomly-chosen images (Fig. 2**b**, purple dots above gray lines, quantile of the median response to preferred images for responses to randomly-chosen images was $q = 0.985$, median across neurons), while the predicted anti-preferred images resulted in responses at the lower bound of these density intervals (Fig. 2**b**, green dots below gray lines, quantile of median response to anti-preferred images for responses to randomly-chosen images was $q = 0.055$, median across neurons). This experimental validation provides clear evidence that V4 neurons have anti-preferred images, identifiable from data-driven DNN models.

A visual neuroscientist may wonder how the responses to anti-preferred images compare with responses to gray, blank screens—the *de facto* stimulus used between stimulus presentations to bring the neurons' firing rates to rest. It is not unreasonable to assume this featureless gray stimulus yields the smallest response. However, we found this not to be the case: While the preferred images strongly drove the time-varying V4 response above baseline (Fig. 2**c**, top, 'preferred'; the response transient is ∼50 ms after 'stim-on' due to synaptic delay from retina to V4), the anti-preferred images suppressed the neuron's response below its baseline firing rate (Fig. 2**c**, top, 'anti-preferred'). We confirmed this across neurons by comparing responses during the 100 ms stimulus presentation (Fig. 2**c**, 'stim.', lagged by +50 ms) versus responses in the 100 ms immediately following stimulus presentation during which a gray, blank screen was presented (Fig. 2**c**, 'blank', also lagged by +50 ms); the results were significant (Fig. 2**c**, bottom) and beyond what we imagined for the anti-preferred images (Fig. 2**c**, bottom right). This rules out the possibility that most anti-preferred images are featureless within a neuron's spatial receptive field. Indeed, the diversity and specificity of model-optimized anti-preferred images is on par with those of model-optimized preferred images (Fig. 2**d**, top versus bottom rows).

## 4 ANTI-PREFERRED STIMULI ARE NEEDED TO ESTIMATE V4 TUNING.

The existence of anti-preferences alone does not necessarily imply that they are essential for shaping a V4 neuron's tuning. Indeed, knowing a neuron's anti-preferred images may not help predict the neuron's responses to other images—a failure to generalize. In contrast, if a neuron prefers a certain visual feature, increasing the strength of this feature in an image is often thought to further drive the response of the neuron (e.g., increasing the contrast of a sinusoidal grating for a V1 neuron). We devised an approach to assess the information content of anti-preferred images by including them or leaving them out when estimating a neuron's tuning, an approach inspired by data pruning (Paul et al., 2021; Sorscher et al., 2022). We chose to estimate the tuning of the data-driven V4 model neurons in place of real V4 neurons, allowing us to consider responses to 500,000 images. We employed different data-pruning strategies by considering different sets of

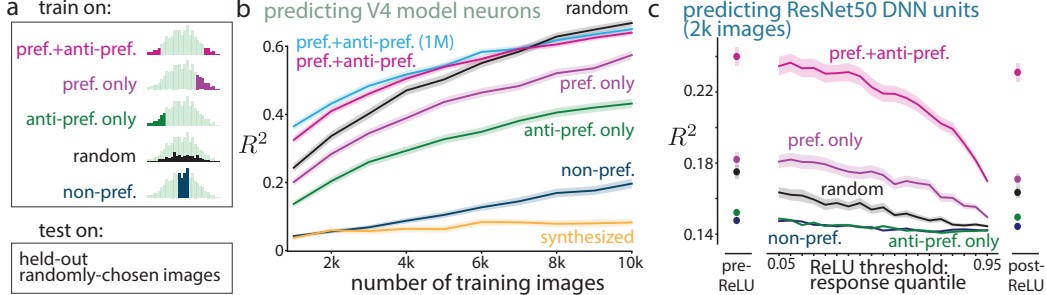

**Figure 3: Anti-preferred images contribute to V4 tuning. a**. Response distributions for different training sets for a data pruning analysis; $R^2$ is always computed with the same held-out natural images. **b**. We train a 5-layer DNN (see Methods A.3) to predict responses of individual V4 model neurons (219 in total), varying the number of training images up to 10k (see Supp. Fig. 2 for results with > 10k images). Response distributions were over 500k images; we also considered a distribution for 1 million (1M) images. **c**. We perform a data pruning analysis for 219 ResNet50 units (2k training images), either pre-ReLU (left) or post-ReLU (right), as well as ReLU thresholds equal to different quantiles (e.g., a quantile of 0.5 results in responses to half the images being above 0). Traces denote means; shaded areas denote 1 s.e.m.

training images: preferred, anti-preferred, both, randomly-chosen, or non-preferred images whose responses were closest to the median response (Fig. 3**a**).

For a small number of training images (<5k images), training on both preferred and anti-preferred outperformed randomly-chosen images (Fig. 3**b**, 'pref.+anti-pref.' versus 'random'). Training either on preferred images alone or anti-preferred images alone failed to reach the prediction performance of random selection (Fig. 3**b**, 'pref. only' and 'anti-pref. only'). Thus, both are needed together to achieve good tuning estimates; consistent with this, training only on non-preferred images had poor generalization (Fig. 3**b**, 'non-pref.'). Choosing more extreme versions of preferred and anti-preferred images (by doubling the pool to 1 million images) led to a further increase in prediction performance (Fig. 3**b**, 'pref.+anti-pref. (1M)'). However, synthesizing images via gradient techniques to maximize and minimize a neuron's response (see Appendix B.3) (Ponce et al., 2019; Bashivan et al., 2019a; Walker et al., 2019; Gu et al., 2022; Cowley et al., 2026; Willeke et al., 2023; Pierzchlewicz et al., 2024) performed poorly (Fig. 3**b**, 'synthesized'), likely because synthesized images deviate too far from the manifold of natural images as well as suffer from a lack of diversity (Pierzchlewicz et al., 2024; Nguyen et al., 2015). We note that, as expected, random-selection eventually outperforms the other biased training sets that are out of distribution (Fig. 3**b**, 'random'). **Nonetheless, these results suggest that by knowing a neuron's preferred *and* anti-preferred images, one can reasonably estimate the rest of the neuron's tuning to other natural images.**

We wondered how these results may change for estimating the tuning of one-tailed DNN units. We performed the same data-pruning analysis on post-ReLU DNN units and found that training only on preferred images performed better than random-selection (Fig. 3**c**, 'post-ReLU'), consistent with the idea that one-tailed units are mostly characterized by their preferred feature. However, we were surpised to find that training on both preferred and anti-preferred images led to a substantial boost in performance (Fig. 3**c**, 'post-ReLU', pink dot). This trend held for the corresponding pre-ReLU units (Fig. 3**c**, 'pre-ReLU') as well as for different ReLU thresholds set to response quantiles (Fig. 3**c**, 'ReLU threshold'). How is it possible for anti-preferred images to help in estimating the tuning for extreme one-tailed units (e.g., response quantile of 0.95)? Consider a unit that detects dogs versus other objects. By only training on images of dogs (preferred images), the tuning model may focus on small response differences to dog breed, size, fur, etc., instead of determining the features/objects—cats, airplanes, fruit, etc.—to which the unit does not respond (its anti-preferred images). This highlights that even for a one-tailed unit, there is benefit in knowing its anti-preferred images (potentially with multiple, diverse features) when estimating its tuning (DiMattina and Zhang, 2008; Cowley and Pillow, 2020).

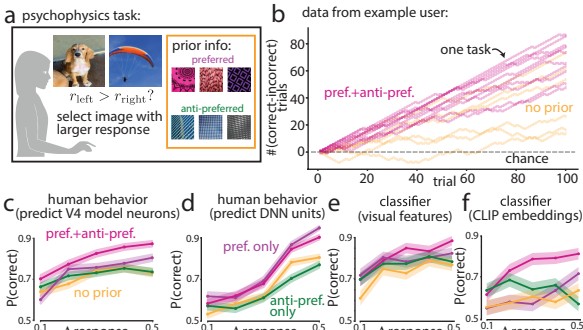

**Figure 4: Humans use anti-preferred images to infer V4 tuning. a**. Psychophysics task. **b**. Performance for one example user with or without prior information. Each trace is the running difference between the number of correct and incorrect choices for one V4 model neuron. **c-d**. Human performance predicting V4 model neurons (**c**) and ResNet50 DNN units (**d**). **e-f**. Task performance in **a** for an online classifier trained on visual features (**e**) or on CLIP embeddings (**f**). Lines: mean, shade: 1 s.e.m.

## 5   HUMANS USE ANTI-PREFERRED IMAGES TO INFER V4 TUNING.

Access to both preferred and anti-preferred images was most informative for estimating V4 tuning (Fig. 3), likely because filters learned to detect the salient visual features in both image types. We wondered to what extent humans could access and interpret these visual features. To test this, we ran a psychophysics experiment in which human subjects chose one of two images that they thought would lead to a larger response of a V4 model neuron (Fig. 4**a**, see Methods A.4); this task was inspired by recent work in explainable AI (Borowski et al., 2020; Zimmermann et al., 2024). Prior to a task, we gave subjects one of four possible sets of reference images: both preferred and anti-preferred images, preferred images only, anti-preferred images only, and no prior images. Subjects improved their performance via feedback of the correct image after each trial (Fig. 4**b**). We tested these four conditions for 10 different V4 model neurons and 10 different DNN units (80 tasks total); we found real V4 responses to be too noisy for humans to predict accurately (see Methods A.4).

The impressive performance of human subjects (Fig. 4**c**, 80.5% mean accuracy) indicates that the visual feature preferences of V4 neurons are distinguishable and human-interpretable. When given prior access to both preferred and anti-preferred images, subjects outperformed other types of prior information (Fig. 4**c**, 'pref.+anti-pref.'); performance for preferred-only and anti-preferred-only was comparable to no prior images (Fig. 4**c**, 'no prior' trace). This suggests that subjects relied on both preferred and anti-preferred images to infer a V4 model neuron's tuning. In contrast, when predicting responses of ResNet50 DNN units, subjects performed similarly when given prior information of either solely preferred images or both preferred and anti-preferred images together (Fig. 4**d**, 'pref.+anti-pref.' versus 'pref. only'), whereas performance dropped for anti-preferred images (Fig. 4**d**, 'anti-pref. only'). Thus, subjects did not rely on the anti-preferred images to predict DNN unit responses, suggesting these images—despite their usefulness in estimating a DNN unit's tuning (Fig. 3**c**)—do not have salient, human-interpretable features. This emphasizes an important difference between V4 neurons and current task-driven DNN models.

What are the visual features that humans use to distinguish between preferred and anti-preferred images? We hypothesized that humans relied on more high-level, interpretable visual features than those typically found in DNN models. To test this, we computed a feature set of 34 interpretable high-level image statistics (contrast, luminance, edge intensity, color, ...) as well as a feature set of 512 embedding variables from the DNN model CLIP (Radford et al., 2021), and used an online preference algorithm (De Peuter et al., 2024) to train a recursive least squares classifier (see Methods A.5) to perform the same psychophysics task to predict V4 model neuron responses. We found better agreement in performance between the interpretable features and human behavior (Fig. 4**e**) than that for the CLIP embeddings (Fig. 4**f**). This supports the idea that there are human-interpretable differences in visual features between preferred and anti-preferred images, which we explore in the next session.

## 6   THE VISUAL FEATURES OF PREFERRED AND ANTI-PREFERRED IMAGES

To assess how visual features differ between preferred and anti-preferred images, we compared 34 high-level, human-interpretable image statistics between the two (Fig. 5**a**, same features as in Fig. 4**d**). For each V4 model neuron, we computed the average difference of each visual feature between its preferred and anti-preferred images (mean over 100 images each). We found large dif-

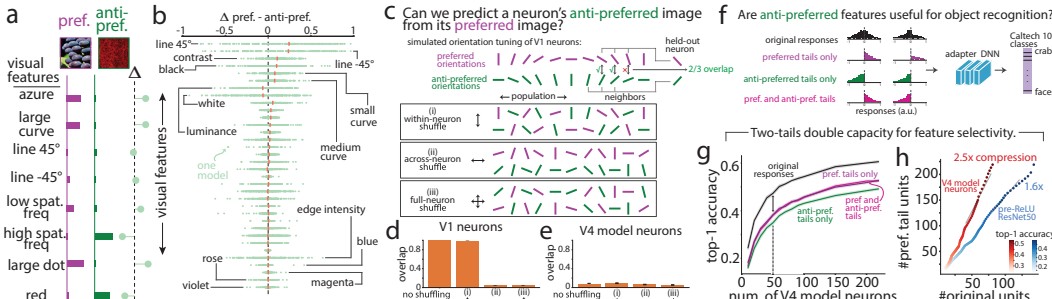

Figure 5: **Anti-preferences double a V4 population's capacity for feature selectivity. a**. Visual features differ between preferred and anti-preferred images. **b**. Mean differences in features that were normalized between 0 and 1. Lines: medians, dots: V4 model neurons. **c**. Simulated preferred and anti-preferred orientations for a population of V1 neurons. Shuffling these preferences breaks specific relationships. **d**. Neighbor overlap for predicting the anti-preferred features using preferred features for simulated V1 neurons for each condition in **c**. **e**. Same as **d** but with V4 model neurons using interpretable visual features. **f**. We assess the extent to which different tailed response distributions perform object recognition. **g**. Top-1 accuracy as a function of number of V4 model neuron responses. **h**. Number of preferred-tail units needed to match accuracy of original (unmodified) units. Traces denote means, shaded areas denote 1 s.e.m.

ferences for individual models (Fig. 5**b**, large variance of dots for each row), indicating that these visual features are capable of differentiating between preferred and anti-preferred images for an individual V4 model neuron. However, we found little overall mean difference in visual features across V4 model neurons (Fig. 5**b**, orange lines close to dashed line), suggesting that a visual feature is as likely to be preferred as anti-preferred. This was surprising, as one may suspect, for example, that high-contrast images would drive larger V4 responses than low-contrast images. While this may be true across randomly-chosen natural images (Sclar et al., 1990), this effect is weak between preferred and anti-preferred images (Fig. 5**b**, 'contrast').

After re-examining the preferred and anti-preferred images (Fig. 2**d**), we wondered if there were any relationships between a neuron's preferred and anti-preferred features. In other words, if two V4 neurons had the same preferred feature, would both neurons also have the same anti-preferred feature? We devised the following analysis and first motivate it with a simulated population of simple V1 neurons, each with its own preferred edge orientation (equally spaced angles, Fig.5**c**, top row). Assuming ideal cosine tuning (Hubel and Wiesel, 1962; Cavanaugh et al., 2002), each neuron's anti-preferred orientation was 90 degrees/orthogonal from its preferred orientation (Fig. 5**c**, top row, purple versus green lines)—a clear relationship. To detect this relationship, we considered a held-out neuron and identified its nearest neighboring neurons (based on angle) for its preferred orientation as well as its anti-preferred orientation. We then compute the overlap in neuron identity between the two sets of neighbors (Fig. 5**c**, top row, 'overlap'). We then shuffle the relationships between preferred and anti-preferred features across the population—either for each neuron individually (within-neuron shuffle), for each feature type individually (across-neuron shuffle), or both (full-neuron shuffle) (Fig. 5**c**, i, ii, and iii). As expected, the V1 neurons had a strong overlap for the non-shuffled population (Fig. 5**d**, 'no shuffling') as well as for the within-neuron shuffled population (Fig. 5**d**, i), but this overlap collapsed when shuffling across neurons (Fig. 5**d**, ii and iii) breaking the relationship of orthogonal orientations for each neuron's preferences. Thus, our overlap analysis can tease apart different relationships between preferred and anti-preferred features.

We performed the same analysis on the V4 model neurons using the Euclidean distance of the 34 interpretable visual features (see Supp. Fig. 8 for results on CLIP embeddings) to identify the nearest neighbors. We found a striking result: overlap was small for all settings, with little difference in overlap between no shuffling and full shuffling (Fig. 5**e**). This was consistent with our finding that a V4 model neuron strongly activated by the preferred images of another model rarely is suppressed by the other model's anti-preferred images (Supp. Fig. 7). Thus, two V4 neurons with similar preferred features will likely have different anti-preferred features. Beyond this, our results suggest the preferred and anti-preferred features of a V4 neuron appear to be randomly and inde-

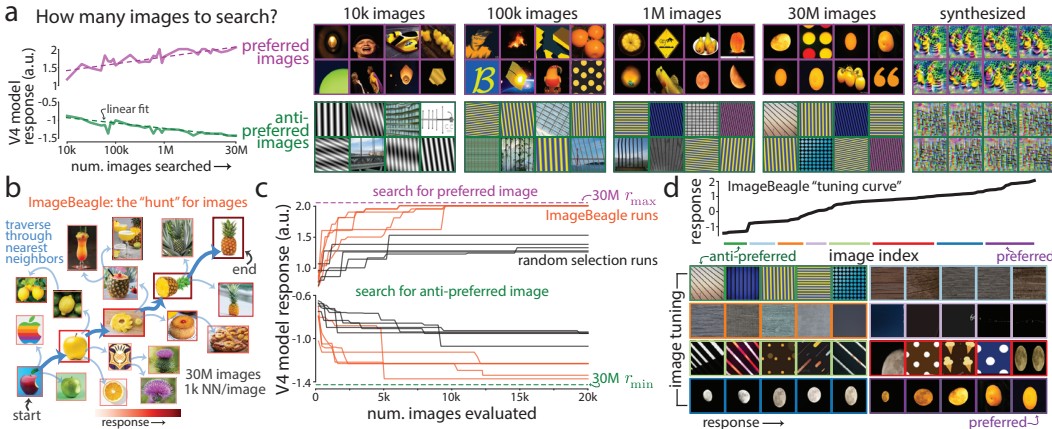

**Figure 6: ImageBeagle searches the natural image manifold to efficiently find preferred and anti-preferred stimuli. a**. A V4 model neuron's responses to preferred and anti-preferred images after searching through a subsample of $K$ images. Dashed-lines: linear fits; $x$-axis is log-scale. Right: Top preferred and anti-preferred images for each $K$ as well as synthesized images. **b**. ImageBeagle navigates the natural image manifold via a nearest neighbor graph. **c**. Runs of Image-Beagle (orange traces) versus random selection (black traces) searching for the preferred (*top*) and anti-preferred (*bottom*) image. Dashed lines: optimal out of 30M images. **d**. ImageBeagle tuning curve for a V4 model neuron.

pendently sampled from the same "preference" distribution. A caveat is that we only considered 219 V4 model neurons here; these results may change as we gain access to more V4 neurons in the future.

## 7 ANTI-PREFERRENCES DOUBLE A V4 POPULATION'S SELECTIVITY.

What is the computational benefit of encoding anti-preferred features? We hypothesize that anti-preferred features allow a population to double its capacity for feature selectivity, providing a more flexible readout for downstream neurons. To test this, we assessed the extent to which the population of V4 model neurons could perform an object recognition task (Caltech-101 (Li et al., 2022)) with or without access to its anti-preferred features (Fig. 5**f**). We trained an adapter DNN to predict the object using either original responses of the V4 model neurons, responses above a ReLU threshold ('pref. tails only'), responses below a ReLU threshold ('anti-pref. tails only'), or a random mixture of both ('pref. and anti-pref. tails'). We found that for the same number of V4 model neurons, the original responses substantially outperformed all other settings (Fig. 5**g**, black trace above others). We quantified this by computing the ratio between the number of preferred-only model neurons divided by the number of original model neurons needed to achieve the same accuracy and found a compression ratio of 2.5 (Fig. 5**h**). Thus, encoding anti-preferred features effectively doubles the feature selectivity of a V4 population, allowing for more informative readout. In comparison, the compression ratio was much lower for pre-ReLU DNN units (Fig. 5**h**, 1.6×), indicating that the anti-preferred features of DNN units were not informative of object identity. This highlights an obvious mismatch between visual cortical neurons and DNN units and suggests a potential way forward to reduce the number of units in DNN models.

## 8 SEARCHING THROUGH MILLIONS OF IMAGES WITH IMAGEBEAGLE

Our work has relied on searching through many natural images to identify preferred and anti-preferred stimuli—how many images are needed? We searched an increasing number of stimuli for a V4 model neuron, and found the number of images had to exponentially scale to achieve a linear increase in response (Fig. 6**a**, left panel). Only at 1 million images did the preferred and anti-preferred images resemble those for 30 million images (Fig. 6**a**, '1M' versus '30M'). A complementary approach is to synthesize images via gradient techniques (Ponce et al., 2019; Bashivan et al., 2019a; Walker et al., 2019; Gu et al., 2022; Cowley et al., 2026; Willeke et al., 2023; Pierzchlewicz et al., 2024) that often identifies images that yield the largest and smallest

responses (for this V4 model neuron, $r_{max}=6.5$, $r_{min}=-3.5$) but can be difficult to interpret versus natural images (Borowski et al., 2020; Geirhos et al., 2021) and are highly stereotyped (Fig. 6**a**, 'synthesized') leading to poor inference of a neuron's tuning (Fig. 3). In our own experiments, we find that natural and synthesized preferred images drive real V4 responses similarly, but that natural anti-preferred images more strongly suppress real V4 responses than those to synthesized anti-preferred images (Supp. Fig. 5). The synthesized approach also requires technical expertise and dedicated hardware that few neuroscience labs have available. This motivated us to design a simple tool for visual neuroscientists and machine learning researchers to efficiently search through millions of natural images to optimize a desired objective (e.g., minimizing a neuron's response).

We developed a new tool, called *ImageBeagle*, that efficiently searches through millions of natural images to "hunt" for a desired stimulus in a short amount of time. The key intuition is that we traverse through the natural image manifold by visiting each image's nearest neighbors, moving to the neighbor with the largest objective value (i.e., a discrete version of gradient ascent, Fig. 6**b**). We collected 30M images from diverse image datasets and computed 1k nearest neighbors for each image. Image similarity was defined as the Euclidean distance between DNN features from ResNet50; we confirmed image neighbors were perceptually similar (Supp. Fig. 3). ImageBeagle alternates between a global search determined by a coreset over images (Bachem et al., 2018) and a local search that moves to the neighbor with the largest objective value (see Methods A.7). We tested the performance of ImageBeagle versus random search and found impressive speed-ups: ImageBeagle often identified preferred and anti-preferred images with resulting responses close to the 30M-optimum after only 10k evaluated images (Fig. 6**c**, orange traces), substantially outperforming random selection (Fig. 6**c**, black traces). We confirmed that ImageBeagle outperformed random selection for models of V1, V4 and IT neurons (Supp. Fig. 3). We also used ImageBeagle to connect a neuron's preferred and anti-preferred images together along a smoothly-varying tuning curve (Fig. 6**d**); smoothness is guaranteed because we constrain ImageBeagle to traverse along the image manifold via nearest neighbors. Such an interpretable tuning curve has been difficult to identify because of the nearly infinite paths possible between two images (Pasupathy and Connor, 2001; David et al., 2006). We suspect ImageBeagle will be of practical value to visual neuroscientists interested in optimizing neurons' responses (Cowley et al., 2017b; Walker et al., 2019; Ponce et al., 2019; Bashivan et al., 2019a), performing closed-loop experiments with active learning (Lewi et al., 2009; Cowley and Pillow, 2020; Cowley et al., 2026), and estimating tuning curves that smoothly vary in stimulus space (Berardino et al., 2017; Wang and Ponce, 2024). Unlike most model-optimized stimuli, ImageBeagle does not require technical expertise, lowering the barrier for adoption by many experimental labs.

## 9 DISCUSSION

Our work establishes the existence and importance of anti-preferred images for stimulus tuning in the visual cortex, especially visual area V4. We systematically investigate the properties of anti-preferred images through experimental validation, modeling, human psychophysics, and image statistics. The existence of anti-preferred images is not obvious: Task-driven DNN units, commonly used to model V4 neurons, often do not exhibit anti-preferences due to their ReLU thresholding. This suggests that a V4 neuron's response less resembles the output of a ReLU and more resembles a filter with a dynamic range. A V4 neuron may form its two-tail selectivity in part by combining excitatory and inhibitory pre-synaptic input from neurons with one-tailed response distributions (Fig. 2**a**) or combining and thresholding activity within its own dendrites (Beniaguev et al., 2021). Our results, consistent with concurrent work optimizing for least activating images of V1 and V4 neurons (Franke et al., 2026), suggest that only characterizing a neuron by its preferred feature misses critical aspects of the neuron's tuning. How two-tailed response distributions and anti-preferred features relate to efficient and sparse coding in the brain (Olshausen et al., 1996; Rozell et al., 2008) remains an open question; two-tailed response distributions may require more energy for spikes, but require fewer neurons to encode rich feature selectivity. To search for anti-preferred images, we developed the ImageBeagle tool that efficiently searches through millions of natural images; ImageBeagle is ready to be used in real-time, closed-loop experiments. Overall, our work marks the beginning of a quest to identify the role anti-preferences play in other biological and artificial visual systems, and improve DNNs inspired by neuroscience principles.

## ACKNOWLEDGMENTS

This work was supported by an NIH grant (F31EY031975) to P.L.S., an NIH grant (R01EY029250) to M.A.S., the Pershing Square Innovation Fund to B.R.C., and an NIH grant (R01EY037194) to M.A.S. and B.R.C.

## DATA ACCESSIBILITY

Our V4 data and code is available at https://github.com/cowleygroup/Gondur_et_al_2026.git. ImageBeagle is available at https://github.com/cowleygroup/ImageBeagle.git.

## BROADER IMPACTS

In our work we establish anti-preferred images as an important property of stimulus tuning in area V4. Although there has been extensive research around preferred images, anti-preferred images have largely been overlooked. Thus, our knowledge of the mid-level processing in the ventral visual stream has been limited by the incomplete picture of the neurons' preferences *and* anti-preferences. We believe this work will inspire experimentalists to investigate the roles of anti-preferred images in visual cortical processing and enrich our understanding of the communication between lower and higher order visual areas. Moreover, with ImageBeagle, experimentalists now will have access to millions of images with a complementary framework that can efficiently search through these images to optimize a general objective function beyond preferred and anti-preferred images (e.g., finding the reddest image).

## LIMITATIONS

One limitation of our work is that for our neural recordings from V4, our results reflect only a small subsample of V4 neurons ($\sim$200 out of millions). Thus, preferred and anti-preferred image features may be better distinguished in overall image statistics with more V4 neurons. Moreover, recorded electrode units were likely a mix of isolated neurons and multi-neuron activity (neural units were spike-sorted with dedicated software (Issar et al., 2020)). If the recordings had multi-neuron activity, possibly one neuron could dominate the rest with its preferred and anti-preferred image responses. Another limitation is that ImageBeagle was partly curated using images from the web and public datasets. We recognize that images scraped from the web can contain a variety of content, some of which might have sensitive information. To minimize the occurrence of this, we sampled from public datasets that use pre-processing steps to eliminate these images. However, similar to any public datasets, we advise users to use ImageBeagle at their own discretion. In addition to this, because computing each image's nearest neighbors out of 30M images can take over a year, we approximated this graph with random sampling, ensuring that nearest neighbors were perceptually similar to the base image. However, this approximation may fail for certain types of images due to a noisy and/or incomplete image graph.

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

# A  METHODS

In Methods, we provide details for our V4 experiments, linear mapping analysis, data pruning analyses, human psychophysics experiment, and ImageBeagle dataset and algorithm.

## A.1  V4 EXPERIMENTAL DATA

Most of our analyses of V4 responses and V4 model neurons come from data collected in our recent work (Cowley et al., 2026); further validation experiments novel to this work were collected during the same recording periods. See Cowley et al. (2026) for full experimental details. These experimental procedures were approved by the Institutional Animal Care and Use Committee of Carnegie Mellon University and were performed in accordance with the United States National Research Council's Guide for the Care and Use of Laboratory Animals.

Specifically, we used this dataset to compute skewness of V4 responses to natural images (Fig. 1**a-c**), predict V4 responses from ResNet50 embeddings with different mappings (Fig. 2**a**), and assess excitation and suppression of preferred and anti-preferred images (Fig. 2**c**). We ran additional closed-loop validation experiments for the anti-preferred images (Fig. 2**b**) interleaved in the recording sessions from Cowley et al. (2026), allowing us to re-use the same compact models of V4 neurons. We also re-used these compact models (termed *V4 model neurons* in our work) as surrogates for real V4 neurons, as we found them to be more predictive of V4 responses than task-driven DNN models (Cowley et al., 2026). We analyzed these V4 model neurons for our data pruning analysis (Fig. 3**b**), psychophysics experiments (Fig. 4**c**), visual feature analyses (Fig. 5), and evaluating ImageBeagle (Fig. 6). Below, we briefly describe data collection of the V4 responses.

*Macaque V4 neural data collection:* We implanted a 96-electrode array in the left hemisphere of macaque visual area V4, one in each of two macaque monkeys (identifiers PE and RA). We extracted spike signals via an automated algorithm (Issar et al., 2020) that separated spike waveforms from noise on each electrode channel. For each recording session, the awake, head-fixed animal performed thousands of active fixation trials until satiated (typically ~2-3 hours). Each trial comprised ~6-8 image flashes (~100 ms each) interleaved with 100 ms gray blank screens (to reduce adaptation and interaction effects between image flashes); image size and location were chosen to cover the receptive fields of the recorded V4 neurons (8-11 visual degrees in diameter at the recorded eccentricity). After maintaining fixation throughout the sequence of images, the central dot disappeared and a target dot appeared at a random location 10° away from the central dot; animals received a liquid reward for correctly making a saccade to the target dot. Each recording session had ~1,000 unique images and typically greater than ~5 repeats per image (image repeats shown randomly throughout the session).

*V4 model neurons:* We used the data-driven compact models trained in our previous work (Cowley et al., 2026) as our 'V4 model neurons'. These compact models were obtained by recording ~10 sessions per animal to train the data-driven model, called the 'ensemble model'. This ensemble model was highly predictive of V4 responses to held-out natural images. The compact models were identified by distilling and pruning the ensemble model; one compact model was identified per V4 neuron. We then fixed the weights of these compact models, and searched for their preferred and anti-preferred images. To do this, we passed ~500,000 natural images through the compact models, and kept the 10 preferred and 10 anti-preferred images for each V4 neuron (for the second animal, we had 12 images each). We then presented these images, along with ~750 randomly-chosen natural images, in the following recording session. Because we could not guarantee that we record from the exact same neurons between sessions (a small number of neurons were lost and added between sessions due to small shifts in electrode positions), we had to match up the V4 model neurons (i.e., the compact models) to the recorded V4 neurons on the new session. To do this, we computed the predicted $R^2$ between each V4 model neuron and each V4 neuron using the responses to the randomly-chosen natural images, and kept greedily choosing the pair with the highest $R^2$ (and removing the chosen model neuron and real neuron as future candidates). Then, for each V4 neuron, we take the median response of the model-optimized preferred images $r_{\text{pref}}$ and anti-preferred images $r_{\text{anti-pref}}$, and compute the fraction/quantile $q$ to which these median responses are either larger (preferred) or smaller (anti-preferred) than responses for the randomly-chosen images, e.g., $q_{\text{pref}} = \frac{1}{N} \sum_{i=1}^{N} \mathcal{I}(r_{\text{pref}} > r_i)$, where $\mathcal{I}$ is the indicator function and $i$ denotes the $i$th image from the $N$ randomly-chosen natural images.

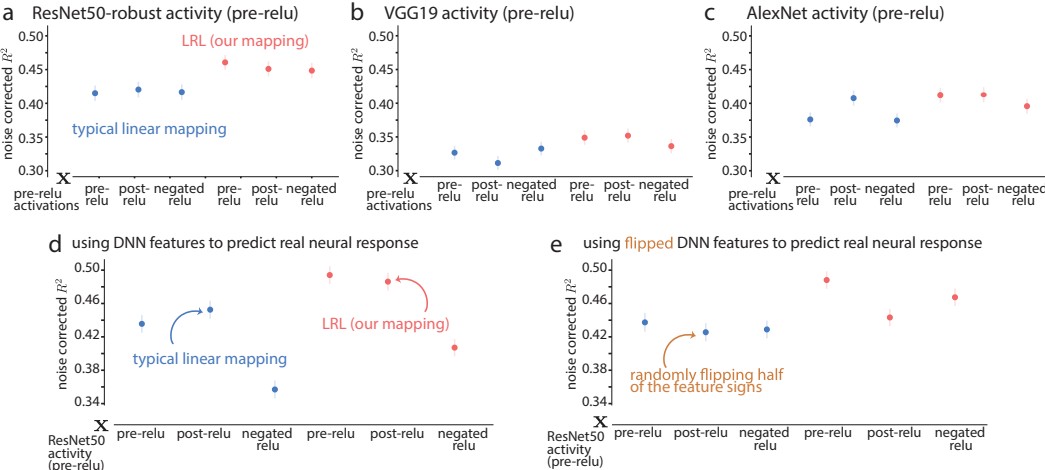

**Supplementary Figure 1: Additional DNNs for linear mapping analysis (*top row*), and flip analysis showcasing the flexibility of LRL mapping (*bottom row*). a**. We extend our mapping analysis from Fig.2**a**. with ResNet50-Robust, **b**. VGG19, **c**. AlexNet. **d**. We repeat our analysis from Fig.2**a**. and **e**. randomly flip the signs of ResNet50 features and try to predict V4 responses for typical and LRL mapping.

## A.2 LINEAR MAPPING ANALYSIS

For our linear mapping analysis (Fig. 2**a**), we used the pre-ReLU activations from ResNet50 ("conv4_block4_add" layer) as our input to all 6 methods, and denote that as "pre-ReLU". For this analysis, we use the four test sessions from Cowley et al. (2026): Session ID 190923, 201025, 210225, and 211022. We keep the same train, test, and validation sets for the final comparisons across all mappings (80% train, 10% test, 10% validation for each session, randomly chosen). Each of the *i-vi* models were the following:

model *i* (pre-ReLU): We linearly map the pre-ReLU features to V4 responses with ridge regression.

model *ii* (post-ReLU): Similar to model *i*, except we add ReLU activation functions prior to the linear mapping. This is the output typically used in identifying linear mappings for V4 neurons (Yamins et al., 2014; Bashivan et al., 2019a; Cowley et al., 2026; Cadena et al., 2024).

model *iii* (negated ReLU): Similar to model *ii*, except we negate the signs of pre-ReLU activations before passing them through the ReLU activation functions.

model *iv*: We linearly combine filter channels via a convolution with kernel shape $1 \times 1$ where the output channels are equal in number to the input channels. The resulting linearly combined activity is first passed through a LayerNorm and then passed through ReLUs until finally linearly mapped to V4 neurons.

model *v*: Similar to model *iv*, except we pass as input post-ReLU activity instead of pre-ReLU activity.

model *vi*: Similar to Method *iv*, except we pass as input negated ReLU activity.

To ensure that our results hold across different DNNs, we repeated this analysis for ResNet50-robust (Engstrom et al., 2019), AlexNet (Krizhevsky et al., 2012), and VGG19 (Simonyan and Zisserman, 2014), and found that all were consistent with our results for ResNet50 (Supp. Fig.1**a-c**). Importantly, one of the advantages of our linear-nonlinear-linear (LRL) mapping is that it is agnostic to the skewness of the responses. In other words, regardless of the two-tailed or one-tailed responses, the LRL mapping can maintain high prediction performance whereas this is not the case with a regular linear mapping (Supp. Fig.1**e**).

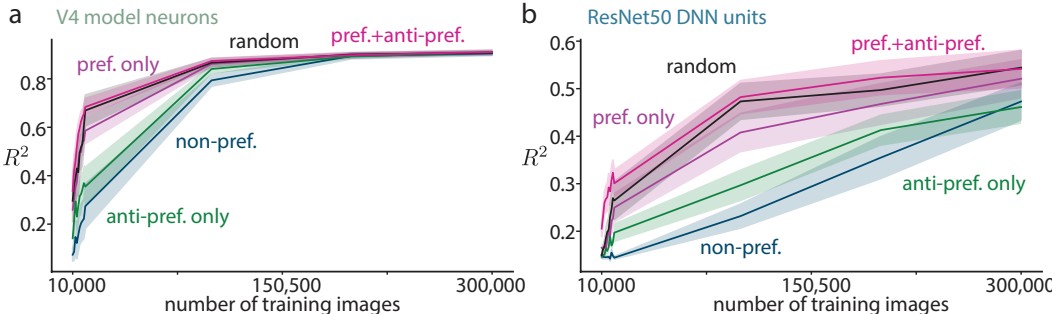

**Supplementary Figure 2: Extended data pruning plots. a**. We train a DNN (5-layer CNN) to predict responses of individual V4 model neurons (10 in total), here with 300,000 training images versus the 10,000 images in Fig.3**b**. Traces denote means, shaded areas denote 1 s.e.m. **b**. Same as in **a** except for predicting responses of individual ResNet50 DNN units (10 in total). Traces denote means, shaded areas denote 1 s.e.m.

### A.3 DATA PRUNING ANALYSES

For our data pruning analyses, we ran simulations to assess the information content of the anti-preferred images by including or leaving out these images when we estimated a neuron's tuning. We used V4 model neurons to serve as surrogate ground truth models of V4 neurons, as recording a real neuron's responses to 500k images is infeasible, and the chosen V4 model neurons were highly predictive of real V4 neurons (Cowley et al., 2026). We then used these surrogates as "teacher" models to train "student" models (5-layer CNN architecture, 100 filters per layer) with different training curricula. We focused on training the student models with less than 10k images, since the most extreme preferred and anti-preferred images likely occur in the first 1k images or so. We trained each model from scratch with training images varying in number from 1k to 10k at 1k intervals (Fig. 3**b**). We used the same teacher-student training procedure for ResNet50 units (Fig. 3**c**). Extending the number of training images did not change the results (Supp. Fig.2). Below we detail the pre-training and training procedures.

*Pre-training details*: Prior to training, for every V4 model neuron, we sorted each model's responses to 500k images; we defined the top $K$ images as the preferred images and the bottom $K$ images as the anti-preferred images. We sought to quantify the extent to which preferred and anti-preferred images contributed to our estimate of that model's tuning. We designed the following different training curricula (corresponding to the traces in Fig. 3**b**), where $K$ refers to the selected number of image-response pairs on which to train.

> *preferred images only:* We selected the $K$ images that had the largest responses.
> *anti-preferred images only:* We selected the $K$ images that had the smallest responses.
> *preferred and anti-preferred images:* We selected $K/2$ images that had the largest responses and $K/2$ images that had the smallest responses.
> *preferred and anti-preferred images (1 million):* We considered an entirely different set of images that was double in size to our baseline dataset (1M here versus 500k for the other curricula). All other details were the same as for 'preferred and anti-preferred images'.
> *randomly-chosen images:* We randomly selected $K$ images from the pool of 500k images.
> *non-preferred images:* We first found the median response and selected the $K$ images with responses closest to the median response (i.e., $K/2$ images with responses below the median response and $K/2$ images with responses above the median response).
> *synthesized images:* We synthesized $K/2$ images to maximize the model's output response and synthesized $K/2$ images to minimize the model's output response. Synthesized images were optimized with gradient ascent/descent techniques (Bashivan et al., 2019a; Walker et al., 2019; Cowley et al., 2026).

The chosen curricula, except for randomly-chosen images, will likely lead to biases such that the training and test data distributions will not match (i.e., out-of-distribution). To mitigate such biases, we replaced 10% of the images for each curriculum with randomly-selected images (replacing the images with the smallest responses for preferred images and the images with the largest

responses for anti-preferred images). We also note that as the number of training images increases, training on randomly-selected images outperforms other curricula as expected, as the training distribution matches the test distribution. Lastly, we added Gaussian noise (with standard deviation 0.2) to the V4 model neuron's responses to mimic noisy estimates of repeat-averaged responses. We used the same training and curricula for predicting ResNet50 units (Fig. 3**c**); we found these units needed more training data than the V4 model neurons to reach similar values of $R^2$ (Supp. Fig. 2).

**Training details**: Across models, the training images were sampled from the same pool of 500k images; these 500k images were randomly sampled from ImageBeagle dataset (see A.6) comprising 30M images. For testing and validation, we sampled 10k images from ImageBeagle for each (images different from the 500k images). We trained the model using the ADAM optimizer with learning rate 1e-4, early stopping (based on validation data), and used a batch size of 8 for ResNet50 and 64 for V4 model neurons.

## A.4 HUMAN PSYCHOPHYSICS EXPERIMENT

We performed a human psychophysics experiment to test how human subjects rely on preferred or anti-preferred images to guess a neuron's (or model unit's) tuning (Fig.4). The subject pool comprised volunteer scientists with no compensation. These experimental procedures were approved by the Cold Spring Harbor Laboratory Institutional Review Board under project "Crack This Neuron" [2282250].

**Task details**: The task was as follows: Given a pair of images, the subject is instructed to select the image that would lead to the larger response of a chosen neuron/model. The user's score and the number of times the user picked the correct answer were displayed above the prompt to show their progress. Importantly, the subject was given feedback after every decision via a green box around the correct answer and red box around the wrong answer. This feedback was necessary for subjects to learn to correctly perform the task. Each task comprised 100 pairs of images. Prior images, if provided, were always 36 images in total; these images were separate from any queried image pairs. To ensure that each task had an equal mix of difficult- and easy-to-discriminate image pairs, we took 5 bins of image pairs with increasing response differences (i.e., the larger the response difference between two images, the easier the discrimination). The first bin had images that were very close in value ($\Delta$ response $\sim 0.1$), and the last bin had images that were very far apart ($\Delta$ response $\sim 0.5$). Each task had 20 image pairs from each bin (100 total) which were randomly ordered across the task. To create these bins, for each model we extracted the responses, and calculated all possible response differences and saved the sorted differences in an array. We then used 20th-80th percentiles of this array to compute bin edges to create our 5 bins and filled these bins with non-overlapping images (e.g., if an image appeared in a pair, it cannot be used for another pair), until each bin had 20 images.

**Task types:** For each task, we used one of 10 units/neurons from either V4 model neurons or ResNet50 units. This created a total of 40 tasks for each model (4 'prior' conditions $\times$ 10 model neurons/units), resulting in 80 tasks total for on subject. For ResNet50, we used 10 randomly selected units from a mid-layer (layer 33, with 1,024 filters). For V4 model neurons, we used 10 randomly selected units from 219 V4 model neurons. In a pilot dataset, we also attempted the task for real V4 neurons, but found the responses too noisy and likely too few images ($\sim$1,000 images per recording session) for humans to identify meaningful selectivity. Our goal was to investigate how humans use prior images (or lack thereof) to guide their decisions. The priors were as follows:

- **no prior:** In this condition, no additional images were shown to the subject. Thus, the subject had to rely heavily on the feedback from the task to guide and improve their decision.

- **preferred prior:** In this condition, we showed the 36 preferred images of the unit. We refer to these images as "maximizing" in the experiment to make it more intuitive for the subjects. Tasks with this condition allow the subject to utilize this prior information by selecting the image from the pair that is most similar to these images.

- **anti-preferred prior:** In this condition, we showed the 36 anti-preferred images of the unit. We refer to these images as "minimizing" in the experiment to make it more intuitive for the subjects. Tasks with this condition allow the subject to utilize this prior informa-

tion by selecting the image from the pair that is not similar to these images. This condition tends to be more challenging compared to preferred prior because here the subject is given information about the lower response images (anti-preferred), but not the higher. Therefore, the user still has to figure out what the maximizing images are through feedback and process-of-elimination strategies.

- **preferred and anti-preferred prior:** In this condition, we showed the user 18 preferred images and 18 anti-preferred images of the model neuron/unit.

## A.5 CLASSIFIER ALGORITHM

We used a recursive least squares (RLS) algorithm (De Peuter et al., 2024) to mimic our human psychophysics experimental setup (Alg.1), where given an image and any prior information, the classifier updates its beliefs through trials. Here, we define the task as a preference learning objective where the classification is based on pairwise feature comparisons and it is sequentially updated based on feedback (e.g. correct/incorrect). RLS algorithm allows for efficient online updates through recursive closed-form solutions, thus aligning closely with the decision making process in the human psychophysics task.

---

**Algorithm 1** Recursive Least Squares Classifier

---

**Require:** Feature vectors $f_t^{pref.}$, $f_t^{anti-pref.}$, target labels $y_t$, forgetting factor $\lambda$, prior type $S$, number of trials $T$

**Require:** Initialize:
  $w$ : weight vector determines the position in the classification hyperplane
  $P = I$ : covariance matrix
  **if** $S \mathrel{!} = $ None **then** initialize $w$ with prior features **else** randomly initialize $w$
  **for** itrial $\leftarrow 1$ to $T$ **do**

  **if** random() $< 0.5$ **then**
    $x_t \leftarrow \frac{f_t^{pref.} - f_t^{anti-pref.}}{\|f_t^{pref.} - f_t^{anti-pref.}\|}$
    $y_t \leftarrow 1$
  **else**
    $x_t \leftarrow \frac{f_t^{anti-pref.} - f_t^{pref.}}{\|f_t^{anti-pref.} - f_t^{pref.}\|}$
    $y_t \leftarrow 0$
  **end if**
  $y_t' \leftarrow 2y_t - 1$
  **Prediction:**
  $s_t \leftarrow w^\top x_t$  # location in the boundary
  $\hat{y}_t \leftarrow \mathbb{I}(s_t > 0)$  # prediction
  **Update:**
  $P_x \leftarrow P x_t$  # uncertainty
  $k_t \leftarrow \frac{P_x}{\lambda + x_t^\top P_x}$  # how strongly should the current trial be weighed
  $e_t \leftarrow y_t' - s_t$
  $w \leftarrow w + k_t e_t$  # classifier weights updated to reduce error
  $P \leftarrow \frac{P - k_t x_t^\top P}{\lambda}$  # updated uncertainty
  **end for**

---

## A.6 IMAGEBEAGLE DATASET AND NEAREST NEIGHBOR GRAPH

ImageBeagle relies on a large bank of millions of diverse images. We collected images from various popular public datasets, such as Flickr Creative Commons dataset (Thomee et al., 2016), Ecoset (Mehrer et al., 2021), and Duckduckgo, in addition to other sources across the web (see Table 1 for approximate number of images extracted from each source). In addition to these, we also created artificial stimuli that are of interest to neuroscience, such as bars, gratings, colorful letters, gaudy images (Cowley and Pillow, 2020), and so on. To ensure consistency across the dataset, we randomly cropped and resized each image to a $224 \times 224$ RGB PNG file. We stored 30 million images in 1,500 zip archive files, where each zip file contained 20k images; we chose zips for easy access and transfer. For easier download and access, we created a miniImageBeagle with 1M im-

ages publicly available to researchers. The full ImageBeagle dataset is large ($\sim$2 TB) and available by request to the authors.

**Nearest neighbor graph:** For our 30 million images, we desired each image's 1k nearest neighbors; an image's neighbors form the local natural image manifold surrounding this image. To obtain the neighbors, we sought the images with the smallest Euclidean distance between activations from a middle layer of ResNet50 (units come from layer 33 of ResNet50). We chose ResNet50 features, as they are predictive of V4 responses (Schrimpf et al., 2018; Cowley et al., 2026), but also as they nicely capture perceptual similarities both for high-level (semantic) and low-level (spatial) statistics (Supp. Fig. 3**a**). We down-sampled the large tensor of activations via a spatial average pool (from $14 \times 14 \times 1,024$ to $3 \times 3 \times 1,024$ with pooling kernel of $4 \times 4$ and a stride of 5). Computing the distance matrix of 30M images is computationally intensive; to mitigate this cost, we approximated these calculations by randomly initializing the 1k nearest neighbors and continuously updating them by computing distances for random subsets of images. We ran this for $\sim$500 hours with H100 GPUs. We confirmed that this similarity metric led to perceptually similar neighbors and effectively searched for preferences for V1, V4, and IT neurons (Supp. Fig. 3**b**).

### A.7 IMAGEBEAGLE SEARCH ALGORITHM

The goal of the ImageBeagle search algorithm is to identify the image that yields the largest value for an objective function given by the user. ImageBeagle consists of 2 steps: Local search and global search. For the local search, we start at a chosen image, compute the objective value for its nearest neighbors, choose the neighbor with the largest objective value, and repeat until the chosen image has an objective value larger than the values of its neighbors. For the global search, we consider $L$ images of a coreset (Sener and Savarese, 2017; Bachem et al., 2018; Kim et al., 2020). A coreset is a subset of images that spreads out over the entire image manifold. Typically, one needs access to the full distance matrix to compute a coreset; here, we approximate the coreset by starting at a randomly-chosen image, and adding to the coreset the image with the largest distance from this chosen image based on ResNet50 features, found by a local search. We repeat this local search (each time searching for the image that maximizes the smallest distance between itself and each of the coreset images) until we have 10k perceptually-dissimilar images. This ensures that we explore diverse regions of the image manifold. Because this coreset is an approximation, we independently computed 10 different coresets for ImageBeagle use.

The full ImageBeagle algorithm (Alg. 2) alternates between the global and local searches to explore the image manifold efficiently for a given objective function. The user defines a budget of the total number of evaluations allowed (i.e., computing the objective value for each image); ImageBeagle stores the objective value for every previously-evaluated image for future queries, ensuring images are not re-evaluated. For the global search, we use the next $L$ images of the coreset and cycle through coresets whenever all 10k images of the current coreset are evaluated. The local search begins using the previously-evaluated image with the largest objective value (and whose nearest neighbors have not yet been evaluated). We perform $M$ local searches (with fresh restarts) before performing another global search. This continues until $B$ images have been evaluated (the budget).

---

**Algorithm 2** ImageBeagle algorithm

---

**Require:** millions of images and 1k nearest neighbors for every image
**Require:** 10 coresets of 10k images each
**Require:**
  **hyperparameters:**
  $L$: number of coreset images searched at each global step
  $K$: number of nearest neighbors to evaluate per local step
  $M$: number of local searches
  $B$: total number of images to evaluate (budget)
**Require:** $\phi(\mathbf{x}) : \mathcal{R}^{p \times p \times 3} \to \mathcal{R}$: objective function with input image $\mathbf{x}$ of shape $p \times p \times 3$
**Set:** num_evals $\leftarrow 0$, $\mathbf{X}_{\text{evaluated}} \leftarrow [\,]$, $\mathrm{X}_{\text{visited}} \leftarrow [\,]$ # empty lists

**while** len($\mathbf{X}_{\text{evaluated}}$) $< B$ **do**

  # **global search**
  $\mathbf{X}_{\text{coreset}} \leftarrow$ next $L$ images in coreset; remove these images from coreset
  **if** coreset empty **then** move to next coreset
  $\mathbf{X}_{\text{evaluated}} \leftarrow [\mathbf{X}_{\text{evaluated}}] + [\mathbf{X}_{\text{coreset}}]$ # combine lists

  # **local search**
  **for** isearch $\leftarrow 1$ to $M$ **do**
    $\mathbf{X}_{\neg\text{visited}} \leftarrow \mathbf{X}_{\text{evaluated}} - \mathbf{X}_{\text{visited}}$
    $\mathbf{x}_{\text{next}} \leftarrow \arg\max(\Phi(\mathbf{X}_{\neg\text{visited}}))$ # choose starting image of local search
    $\mathbf{X}_{\text{visited}} \leftarrow [\mathbf{X}_{\text{visited}}] + [\mathbf{x}_{\text{next}}]$
    $\mathbf{X}_{\text{nearest neighbors}} \leftarrow K$ nearest neighbors of $\mathbf{x}_{\text{next}}$

    # walk through nearest neighbors
    **while** $\max(\Phi(\mathbf{X}_{\text{nearest neighbors}})) > \phi(\mathbf{x}_{\text{next}})$ **do**
      $\mathbf{X}_{\text{evaluated}} \leftarrow [\mathbf{X}_{\text{evaluated}}] + [\mathbf{X}_{\text{nearest neighbors}}]$ # neighbors have been evaluated
      $\mathbf{x}_{\text{next}} \leftarrow \arg\max(\Phi(\mathbf{X}_{\text{nearest neighbors}}))$
      $\mathbf{X}_{\text{visited}} \leftarrow [\mathbf{X}_{\text{visited}}] + [\mathbf{x}_{\text{next}}]$
      $\mathbf{X}_{\text{nearest neighbors}} \leftarrow K$ nearest neighbors of $\mathbf{x}_{\text{next}}$
    **end while**
  **end for**
**end while**
$\mathbf{x}_{\text{optimal}} \leftarrow \arg\max[\Phi(\mathbf{X}_{\text{evaluated}})]$

---

| ImageBeagle | |
|---|---|
| Source | Approximate Amount |
| Flickr Creative Commons dataset Thomee et al. (2016) | 7 million |
| Ecoset Mehrer et al. (2021) | 1.5 million |
| CIFAR Krizhevsky et al. (2009) | 120,000 |
| CelebA dataset Liu et al. (2015) | 202,000 |
| Caltech-256 dataset Griffin et al. (2007) | 30,000 |
| Fashion MNIST dataset Xiao et al. (2017) | 60,000 |
| SVHN dataset Netzer et al. (2011) | 248,000 |
| Google Landmarks dataset Weyand et al. (2020) | 4.1 million |
| DiffusionDB Wang et al. (2022) | 5 million |
| Duckduckgo | 155,000 |
| Flickr | 2 million |
| YouTube | 1 million |
| Artificial stimuli | 3.1 million |
| Others | 5.5 million |

Table 1: Summary of image sources for ImageBeagle

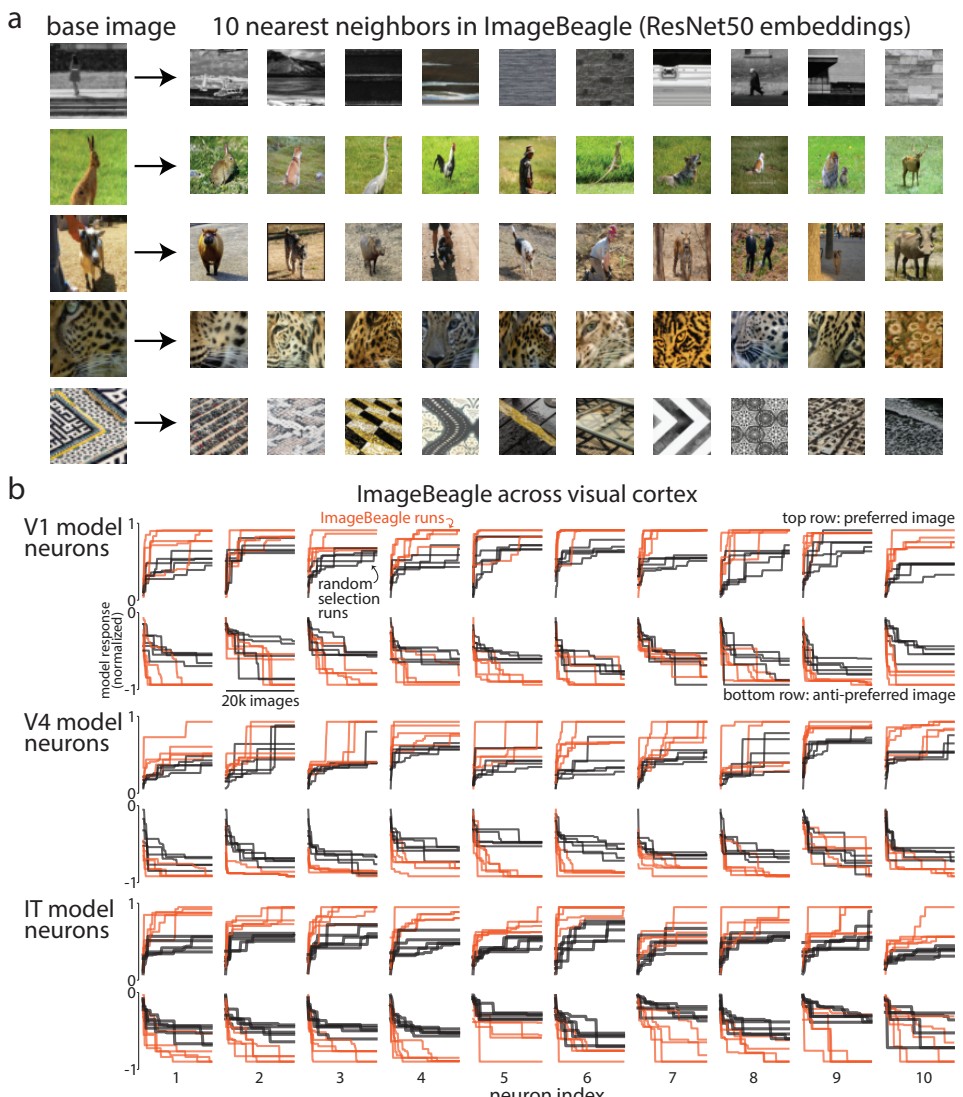

**Supplementary Figure 3: Example ImageBeagle neighbors and searching for preferred and anti-preferred images for neurons from areas across visual cortex. a**. Randomly-chosen example 'base' images and their 10 nearest neighbors based on distances of embeddings from our chosen DNN (ResNet50). **b**. Each base image and its neighbors (**a**) are perceptually similar in low-level statistics (textures, colors, ...), allowing ImageBeagle to be useful in identifying preferred and anti-preferred images for neurons in different visual cortical areas (e.g., V1, V4, and IT) that may differ in receptive field size. To test this, we considered computational models of neurons from V1, V4, and IT (compact models from Cowley et al. (2026)); the V4 and IT models are currently some of the most predictive models of V4 and IT neurons for their respective datasets on BrainScore (Schrimpf et al., 2018). For each model neuron, we used ImageBeagle to identify the preferred and anti-preferred image (Supp. Fig. 3**b**, orange traces), comparing this search with random selection (Supp. Fig. 3**b**, black traces). We found ImageBeagle to perform remarkably well across visual cortical areas (Supp. Fig. 3**b**, all orange traces above black traces in top rows and below black traces in bottom rows), indicating that ImageBeagle often identified images that lead to more extreme responses with fewer image evaluations than those of random selection. That ImageBeagle performs well across visual areas likely stems from the ability of ResNet50 embeddings to identify perceptually similar images (**a**). This suggests that ImageBeagle goes beyond V4 and generalizes to visual neurons in both lower- and higher-order visual areas.

# B  APPENDIX

## B.1  MULTI-UNIT ACTIVITY AND ANTI-PREFERRED IMAGES

To record neurons in our experiments, we used a Utah multi-electrode array, which captures the activity of both single- and multi-units. Identifying well-isolated single units by analyzing spike waveforms is possible, but one can never be certain the unit is truly a single neuron in this setting. Here, we argue that multi-unit activity cannot largely explain the existence of anti-preferred images. This is for two reasons:

First, we analyzed a separate dataset of V4 responses to natural images from Cadena et al. (2024) that used NeuroPixel probes (NeuroNexus V1x32-Edge-10mm-60–177) to record neural activity. An advantage of NeuroPixels is that the electrode channels are much closer together ($50\mu m$) than those of the Utah array ($400\mu m$); one can isolate single units based on coincidence timings of spikes between channels. The authors also performed extensive spike sorting to ensure well-isolated single units. A caveat is that images were presented with only one repeat and in rapid succession without interleaved blank screens—thus, responses are considerably noisier than our analyzed repeat-averaged V4 responses. We kept neurons with a SNR of at least 0.5 (split-repeat analysis). The median skewness of the Cadena dataset was $\kappa = 1.377$ (mean firing rate = 8.76 spikes/sec). We found a tight relationship between mean firing rate and skewness: For neurons with firing rates > 10 spikes/sec, the skewness was $\kappa = 1.06$, and for neurons with firing rates > 15 spikes/sec (similar to our analyzed V4 data), the skewness was 0.852, matching closely to our observed $\kappa = 0.87$ (Fig. 1**c**). That neurons with lower firing rates had higher skewness is unsurprising in this dataset due to the Poisson nature of spike counts. Because each image was repeated a single time, estimating the true response distribution is not possible. However, we believe that the low skewness of the well-isolated high-firing neurons is interesting, as this need not be the case. That the skewness values match well with our observed ones when controlling for firing rate further confirms the presence of two-tailed response distributions for V4 neurons.

Second, because multi-unit activity is thought to be additive, it is likely the case that the image that maximizes the response of the multi-unit likely maximizes an individual neuron within the multiple neurons that comprise the multi-unit. This assumption is often made by recent studies that optimize preferred stimuli of V4 and IT neurons (Bashivan et al., 2019a; Ponce et al., 2019; Cowley et al., 2026; Pierzchlewicz et al., 2024). With similar logic, assuming each unit has an anti-preferred stimulus, identifying the anti-preferred stimulus of a multi-unit is akin to minimizing the response of one of the individual neurons. However, if the units had one-tailed distributions, adding the responses of enough of these units together would likely yield a Gaussian distribution (based on the central limit theorem). To test how many units would need to be added together, we added DNN units with larger skewness ($\kappa \sim 5$), and found that we needed $\sim 400$ DNN units to match the skewness of observed V4 neurons ($\kappa = 0.87$)—this is unrealistic for a real multi-unit, which likely comprises only two or three neurons. The summed responses of three DNN units (over 100 runs) yielded an average skewness of $\kappa = 1.87$, much larger than the skewness for V4. We found that the anti-preferred images for the multi-units of 3 DNN units and above did not have discernible shared visual features versus the perceptually-similar anti-preferred images of the real V4 neurons.

## B.2  SKEWNESS FOR TASK-DRIVEN DNN UNITS

In Figure 1, we considered the skewness of one task-driven DNN (ResNet50). Here, we compare the skewness of four task-driven DNNs to that of V4 neurons (Supp. Fig. 4). Specifically, we wanted to investigate how the highly predictive ReLU layers of popular DNNs organized their responses compared to ResNet50. To do this, we measured the population skewness from 4 popular DNNs that are known to be predictive of V4 responses, Xception (Chollet, 2017) (728 units, from 'block10_sepconv1_act' layer), AlexNet (Krizhevsky et al., 2012) (384 units, from 'conv3' layer), VGG-19 (Simonyan and Zisserman, 2014) (512, from 'block5_conv2' layer), and NASNet-Mobile (Zoph et al., 2018) (528 units, from 'activation_104' layer) (Supp. Fig. 4**a**.-**d**.). While units from AlexNet, Xception, and VGG-19 overall appear to be more skewed than those of V4 neurons (median $\kappa = 2.02$, median $\kappa = 0.94$, median $\kappa = 1.34$ respectively), surprisingly for NASNetMobile (median $\kappa = 0.44$) we found that to be not the case (Supp. Fig. 4**d**). Despite the ReLU activation, units in NASNetMobile organize the responses in a way that preserves the two-tailedness of the distribution, and thus effectively exhibit a linear/pre-ReLU behavior. We suspect that this is caused by the high baseline activations where these units rarely operate in the zero-output regime, hence

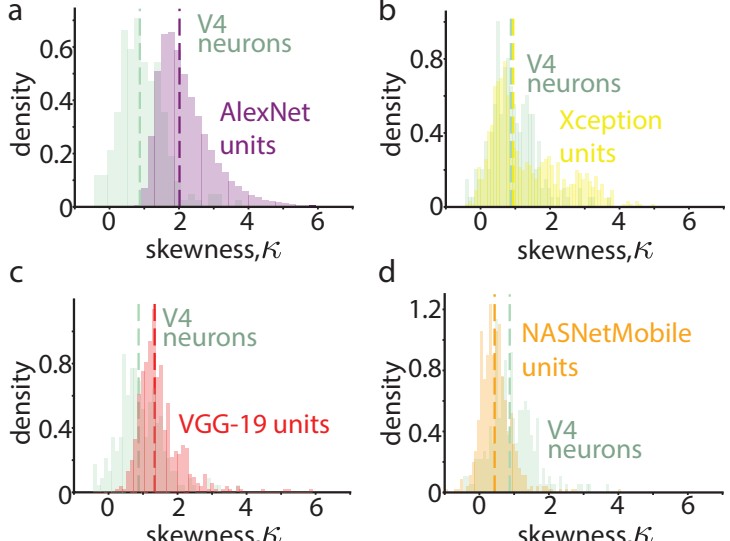

**Supplementary Figure 4: Skewness $\kappa$ of V4 neurons against four different DNNs.** **a**. Skewness $\kappa$ of response distributions for V4 neurons from Fig.1**c**. and AlexNet units. **b**. Same as **a** but with Xception units. **c**. Same as **a** but with VGG-19 units. **d**. Same as **a** but with Nas-NetMobile units. Lines: medians.

creating two-tails. However, thresholding is still present in the model, as a nontrivial fraction of activations still fall below zero in some layers. This finding hints at the importance and effects of architectural designs of DNNs in their selectivity. Compared to ResNet50, AlexNet, VGG-19, and Xception, NASNetMobile utilizes modular cell structure where each cell combines the outputs from previous layers with 'add' operations, thus accumulating the activations from multiple layers. This accumulation of activation increases the baseline activations, leading to ReLU outputs that are always "on". Although the residual skip connections are present in other DNNs such as ResNet50, we suspect the frequency of these additions in NASNetMobile is what leads them to have higher cumulative pre-activations.

### B.3 COMPARING THE EFFECTIVENESS OF MODEL-OPTIMIZED NATURAL VERSUS SYNTHETIC IMAGES

The two main approaches to identify preferred images of V4 and IT neurons have been either to consider a large number of natural images (Cowley et al., 2017a; 2026; Willeke et al., 2023; Wang et al., 2024) or synthesize images via gradient-techniques (Bashivan et al., 2019a; Ponce et al., 2019; Cowley et al., 2026; Willeke et al., 2023; Pierzchlewicz et al., 2024). In our closed-loop experiments (Fig. 2**b**), we had the opportunity to compare the effectiveness of both approaches. Before the validation experiment, we optimized the responses of V4 model neurons by searching for their preferred images either by a large pool of 500,000 natural images (Supp. Fig. 5**a**, 'chosen by model from natural images') or by synthesizing the image via gradient techniques (Supp. Fig. 5**a**, 'synthesized by model'). We then presented both sets of images in a following recording session and compared the V4 responses to each, after establishing which model neurons matched which V4 neurons (see Methods Section A.1). We found no significant difference between model-synthesized and model-chosen images for preferred images (Supp. Fig. 5**a**, bottom), despite salient differences in image statistics between the two sets (Supp. Fig. 5**a**, top), suggesting both approaches are comparable in eliciting large responses and share common visual features. However, for the anti-preferred images (Supp. Fig. 5**b**, top), we found that the model-chosen natural images elicited responses significantly *smaller* than those for the model-synthesized images (Supp. Fig. 5**b**, bottom). This suggests that there is a model mismatch between its predicted anti-preferred images versus the real neuron's anti-preferred images; how visual features differ between these model-chosen natural images and the model-synthesized images may provide insight into these differences. This result also highlights that one should not disregard natural images for their ability to drive both large and small V4 responses. Further work is needed to systematically compare different approaches for pool-based and synthesized-based approaches for identifying both preferred and anti-preferred images for different visual cortical areas.

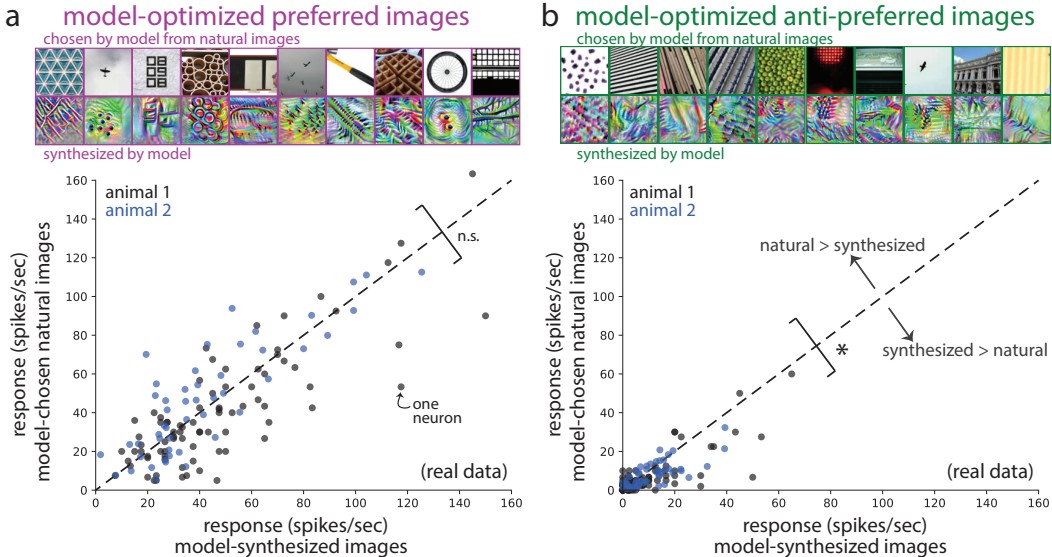

**Supplementary Figure 5: Experimental validation of model-chosen natural images and model-synthesized images. a**. Top panel: Example response-maximizing images either chosen by the V4 model neurons from a large pool of 500,000 natural images (top row) or synthesized by the model via gradient techniques (bottom row). After optimizing these images, we then experimentally validated them by recording V4 responses to these images in a following recording session. Bottom panel: Real V4 responses to the predicted preferred images by the V4 model neurons. Responses were repeat-averaged. Each dot denotes one neuron; neurons are from two animals, indicated by color. The dashed unit line denotes equal responses between model-chosen and model-synthesized images. The difference in means was not significant ('n.s.', $p = 0.514$, two-tailed permutation test, $n = 130$ neurons), suggesting neither approach elicited larger responses than the other. **b**. Same as in **a** except for anti-preferred images. The difference of means was significant (asterisk, $p < 0.02$, two-tailed permutation test, $n = 130$ neurons), indicating that the responses to model-chosen natural images were smaller than those for the model-synthesized images (i.e., dots below dashed line).

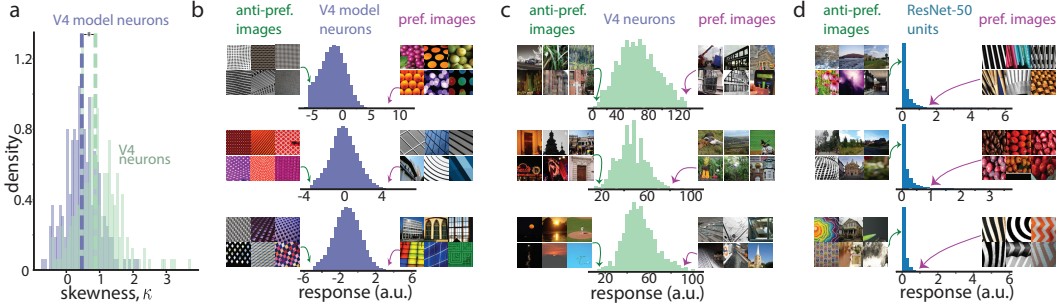

**Supplementary Figure 6: Additional examples of preferred and anti-preferred images from 3 ResNet50 units and V4 model neurons. a**. Skewness of response distributions for V4 neurons and V4 model neurons. **b**. Anti-preferred and preferred images of 3 V4 model neurons. **c**. Anti-preferred and preferred images of 3 ResNet50 units.

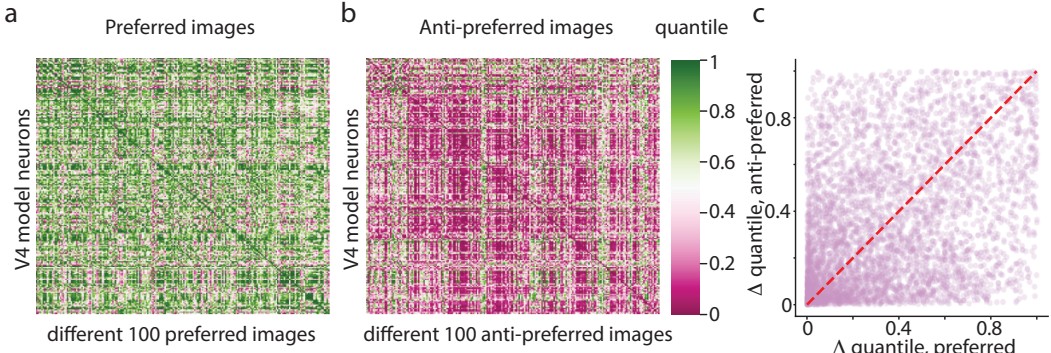

**Supplementary Figure 7: Preferred images are not shared across V4 model neurons. a**. Responses of every V4 model neuron (rows) to every other V4 model neurons' 100 preferred images (columns). **b**. Responses of every V4 model neurons to every other V4 model neurons' 100 anti-preferred images. The mixture of pink and green indicates that some anti-preferred images were close to being preferred images of other models. **c**. Differences in quantiles for the preferred images matrix, and their corresponding differences in quantiles in the anti-preferred images matrix. The dashed red line represents the unity line (y=x).

### B.4 LITTLE RELATIONSHIP BETWEEN PREFERRED IMAGE SIMILARITY AND ANTI-PREFERRED IMAGE SIMILARITY ACROSS MODELS.

To further investigate whether there is shared structure between preferred and anti-preferred images for a given V4 model neuron, we fed the sets of 100 preferred images of all V4 model neurons, and recorded the responses (we repeated this process for anti-preferred images as well). In order to normalize responses across all V4 model neurons, we scaled the responses proportional to the model's true preferred/anti-preferred images by using quantiles. For each set of preferred/anti-preferred images, we took the median and computed how many images out of the 500k had responses lower than this median. We normalized this number by the total number of images to get the quantile response.

Here, a quantile of 1 indicates that the preferred images of the $i$th model is also the preferred image of the $j$th model, and a quantile of 0 indicates that the anti-preferred image of the $i$th model is also the anti-preferred of the $j$th model. We further assessed whether two models with similar preferred images would also have similar anti-preferred images. For every row in Supp. Fig.7**a**, we calculated the absolute difference of quantiles between the model with the highest quantile and the model with the second highest quantile. We did this for the farthest quantiles (highest quantile - lowest quantile) and 20 randomly-chosen quantiles (highest quantile - randomly selected quantiles). We computed the corresponding models' differences from the anti-preferred matrix and compared these $\Delta q$'s against each other. Here, we find that although some models have similar preferred and anti-preferred images (lower bottom left corner in Supp. Fig.7**c**), others do not (Supp. Fig.7**c**, top left corner, bottom right corner). Overall, most images did not fall on the unity line indicating that there is no linear relationship. These results further support our findings of no apparent structure shared across preferred and anti-preferred images (Fig. 5).

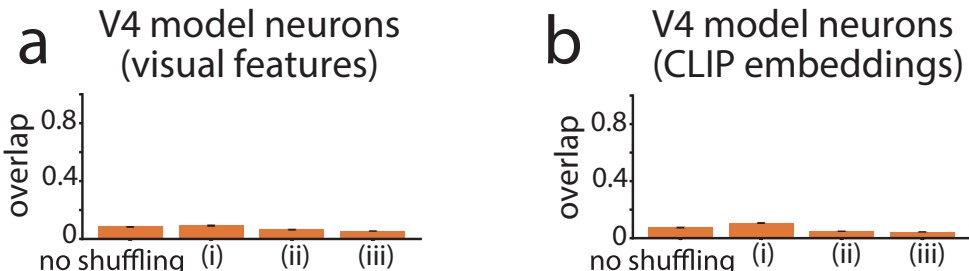

**Supplementary Figure 8: Predicting anti-preferred images based on nearest neighbors of preferred images. a**. Neighbor overlap for predicting the anti-preferred features using preferred features with V4 model neurons using 34-dim interpretable features. **b**. Same as **a** but using 512-dim CLIP embeddings instead.

