# OpenReview forum: "A tale of two tails: Preferred and anti-preferred natural stimuli in visual cortex"
_ICLR.cc/2026/Conference — ICLR 2026 Poster_

### Official Review · Reviewer_c1B5 · 2025-10-27

**Soundness:** 3
**Presentation:** 3
**Contribution:** 3
**Rating:** 8
**Confidence:** 4

**Summary:**

This paper studies the representation of “anti-preferred” visual stimuli—images that suppress neural activity rather than increase it—in the macaque visual system. Using electrophysiological recordings, data-driven modeling, image synthesis, and human psychophysics, the authors report that many macaque V4 neurons show two-tailed response distributions, whereas units in standard deep neural networks (DNNs) display one-tailed, ReLU-like activation patterns. This indicates that cortical neurons can encode information through both excitation and suppression, while DNN units represent information only through positive activations. The study also proposes a modified readout that combines pre-ReLU features before a nonlinearity, improving predictions of neural responses. The manuscript is clearly written, well organized, and presents high-quality figures.

The paper’s main strengths are its clear presentation, strong experimental design, and integration of neuroscience and computational modeling to address an important question at the intersection of biological and artificial vision. The main weaknesses concern the fairness of the readout comparison, the interpretation of the pruning and psychophysics analyses, and limited discussion of prior and related work, as well as the lack of broader connections to artificial vision systems.

**Strengths:**

The study is technically sound and clearly presented. It addresses an important question at the intersection of neuroscience and machine learning by contrasting biological and artificial visual representations. The results suggest that two-tailed neural tuning may provide a richer representational basis than the strictly positive codes used in DNNs. The experiments are well executed, and the integration of electrophysiology, computational modeling, and psychophysics is well thought out. The figures are clear and effectively illustrate the main results.

**Weaknesses:**

The comparison between the proposed readout and the standard linear mapping could be made clearer. While the authors compare against standard post-ReLU readouts, it is not fully clear whether a capacity-matched baseline—applying the same operations (1×1 mixing, LayerNorm, ReLU) after the ReLU—was included. Such a control would help determine whether the observed advantage truly reflects the benefit of pre-ReLU mixing, which allows access to both positive and negative features, rather than increased model flexibility due to the additional operations. Including or explicitly discussing this control would make the interpretation more convincing.

The image synthesis procedure is described clearly, but the discussion could address the limitations of optimizing stimuli far from the natural image distribution. Several recent approaches, such as diffusion-guided activation maximization or Fourier-phase–constrained optimization, maintain proximity to the natural image manifold while exploring feature space. Citing and briefly discussing such work would situate the present synthesis method within the broader literature.

The data-pruning analysis requires a more careful interpretation. The authors conclude that including anti-preferred images is essential for estimating neural tuning, but this improvement may result from greater coverage of the response range rather than from the special status of anti-preferred images. For neurons with sparse selectivity, non-preferred images still provide valuable contrast for estimating tuning, even if there are no distinct anti-preferences. Acknowledging this possibility would clarify the mechanism underlying the observed effect.

Similarly, the psychophysics results are informative but should be interpreted with caution. It is unclear whether the benefit of including anti-preferred images was specific to neurons with two-tailed response distributions or whether any inclusion of low-response examples helps learning. The results from DNN simulations suggest that the latter could be true. If so, the human data may demonstrate a general advantage of diverse examples rather than direct evidence for two-tailed encoding. Rewording this section to reflect this distinction would improve clarity.

The work would also benefit from a more complete engagement with prior neuroscience literature. Selective suppression has been described previously in both V1 and V4. In addition, a recent preprint (bioRxiv: 2025.07.16.665209) reports closely related findings on the role of suppressive features in shaping visual representations. Discussing this work and clarifying how the present study differs in methodology and conclusions would strengthen the positioning of the paper within the current literature.

Finally, given that this paper is submitted to ICLR, the discussion could better connect the findings to implications for artificial vision systems. Two-tailed selectivity may suggest useful design principles for artificial models, such as incorporating balanced excitatory and suppressive representations to improve robustness or generalization. A short discussion of such implications would make the paper more relevant to the machine learning community.

**Questions:**

Was a capacity-matched control readout tested—for example, a variant applying the same mixing and normalization operations after the ReLU—to isolate the specific contribution of pre-ReLU access to negative features?

Can the authors clarify whether the psychophysics effects were stronger for neurons with two-tailed response distributions, or whether similar effects occurred for sparse or one-tailed neurons as well?

How robust are the findings to the specific image synthesis method used? Would on-manifold synthesis (e.g., diffusion-guided optimization) produce comparable preferred and anti-preferred examples?

How do the authors view the implications of their results for representation learning in artificial systems? Could introducing bidirectional or opponent-like feature tuning improve generalization or interpretability in DNNs?

Could the authors comment on how their findings relate to the recent bioRxiv preprint (2025.07.16.665209) reporting suppressive feature selectivity in visual cortex?

---

> ### Author Response · Authors · 2025-11-21
>
> We thank the Reviewer for recognizing that our work addresses an important question at the intersection of neuroscience and machine learning. Below we address the questions and concerns raised by the Reviewer.
>
> > The comparison between the proposed readout and the standard linear mapping could be made clearer. [...]
>
> This is a great point. What inspired us to do our analysis in Fig. 2a was the fact that we know that real V4 responses have two-tails (Fig.1), yet when we look at the most predictive layer of V4 responses in a DNN, we find that they are one-tailed (Fig. 2a, *ii*). This motivated us to check the pre-ReLU layer right before the most predictive layer, which exhibits two-tails similar to V4 responses. We found that this pre-ReLU layer performed significantly worse at predicting V4 responses than the post-ReLU layer (Fig. 2a, *I* vs. *ii*), this is surprising, because the post-ReLU layer has less stimulus information by thresholding the activity. We hypothesized that  this difference in performance is due to the fact that the post-ReLU layer creates a bank of one-tailed responses, which the linear mapping combines to form two-tailed response distributions to match the V4 responses, linearly combining two-tailed responses from the pre-ReLU layer would only work if both the preferred and anti-preferred features of individual pre-ReLU units matched that of individual V4 neurons. To test this hypothesis, we considered the linear-relu-linear mapping that is able to “splice” the two-tailed features of the pre-ReLU activity and stitch them together. This mapping achieves prediction performance higher than the post-ReLU activity (Fig. 2a, *v* vs. *ii*), suggesting that access to both preferred and anti-preferred features better aligns with V4 responses. We have now tried the same linear-relu-linear mapping but with post-ReLU activity, as suggested by the Reviewer, and find a noise-corrected $R^2$ = 0.478, similar to our prediction performance for the linear-relu-linear mapping of pre-ReLU activity (Fig. 2a, *v*, $R^2$ = 0.483). If the improvement was solely due to an increase in nonlinear capacity, we would expect the post-ReLU linear-relu-linear mapping to significantly outperform the pre-ReLU linear-relu-linear mapping, much like we see for ridge regression (Fig. 2a, *i* vs. *ii*). This also hints that the additional non-linearities in the capacity-matched model is not necessary for prediction. Taken together, these results explain why pre-ReLU activity is less predictive than post-ReLU activity up to a linear mapping despite the post-ReLU activity having strictly less information, the pre-ReLU’s preferred and anti-preferred two-tails are poor matches for those found in V4 neurons. If we allow the mapping access to individual preferred and anti-preferred features of the pre-ReLU activity, pre-ReLU and post-ReLU activity lead to similar prediction performance. Thus, the layer with the best prediction, up to a linear mapping, may be misleading, it is not the case that the post-ReLU activity is more “aligned” to the V4 responses, their one-tailed response distributions do not match those of real V4 neurons (Fig. 1). Our linear-relu-linear mapping corrects for a mismatch in preferred and anti-preferred features. In this case, we find that the anti-preferred features are poor predictors of V4 responses (ridge regression on relu(-pre-ReLU activity), $R^2$ =0.347); however, our linear-relu-linear mapping on pre-ReLU activity will outperform the same mapping on post-ReLU activity if the anti-preferred features are better matches to the anti-preferred features of V4 responses. Thus, for predicting neurons with two-tailed response distributions, our linear-relu-linear mapping is agnostic to whether the inputs are two-tailed or one-tailed. Overall, these modeling results provide not only a more predictive mapping but also more intuition of why post-ReLU layers tend to be more predictive of V4 responses. We now rephrase this section and add these additional results to the figure, and include a discussion of this in the revised manuscript. We now also explore this in the new analysis with downstream decoding for an object recognition task.
>
> > The image synthesis procedure is described clearly, but the discussion could address the limitations of optimizing stimuli far from the natural image distribution. [...]
>
> We thank the Reviewer for these suggestions, we certainly will add these to our discussion of model optimized stimuli.
>
> We address further points below.

---

> > ### Author Response · Authors · 2025-11-21
> > **Cont.**
> >
> > > The data-pruning analysis requires a more careful interpretation. [...]
> >
> > We performed our data pruning analysis (Fig. 3a-b) in V4 model neurons as well as on DNN units (Fig. 3c), which indicates that preferred images alone (which are likely more extreme than the weaker responses to randomly-chosen images) outperform randomly-chosen images in this setting. In addition, the analysis in Figure 3 is to verify that both preferred and anti-preferred images are enough to capture a V4 neuron’s tuning (with the obvious caveat that with enough data, randomly-choosing images trumps most data pruning strategies). This analysis indicates that anti-preferred images contribute to a V4 neuron’s tuning. Moreover, humans use these anti-preferred images along with preferred images to estimate a neuron’s tuning (Fig.4b), we find both of these together to be a strong support for the contribution of anti-preferred images.
> >
> > > Similarly, the psychophysics results are informative but should be interpreted with caution. It is unclear whether the benefit of including anti-preferred images was specific to neurons with two-tailed response distributions or whether any inclusion of low-response examples helps learning. The results from DNN simulations suggest that the latter could be true. If so, the human data may demonstrate a general advantage of diverse examples rather than direct evidence for two-tailed encoding. Rewording this section to reflect this distinction would improve clarity. The work would also benefit from a more complete engagement with prior neuroscience literature. Selective suppression has been described previously in both V1 and V4. In addition, a recent preprint (bioRxiv: 2025.07.16.665209) reports closely related findings on the role of suppressive features in shaping visual representations. Discussing this work and clarifying how the present study differs in methodology and conclusions would strengthen the positioning of the paper within the current literature.
> >
> > We thank the reviewer for their suggestions. Due to limited space, we kept the discussion of the prior work brief but we agree that this could be improved. We thank the reviewer for suggesting a concurrent relevant recent work (Franke et al., 2025) on this area, we will revise the final manuscript to include this work. It is great to see separate labs finding the same principles, the field is on to something!
> >
> > For our psychophysics experiment, we specifically included V4 model neurons with two-tails to assess if humans use both of the tails to infer a neuron’s tuning. When given the preferred and anti-preferred images, users performed much better in V4 model neurons compared to DNN units (where anti-preferences are not structured). We believe this shows that despite the response diversity, humans can pick on some structure in anti-preferred images to aid in their estimation.
> >
> > > Finally, given that this paper is submitted to ICLR, the discussion could better connect the findings to implications for artificial vision systems. Two-tailed selectivity may suggest useful design principles for artificial models, such as incorporating balanced excitatory and suppressive representations to improve robustness or generalization. A short discussion of such implications would make the paper more relevant to the machine learning community.
> >
> > To make our contributions to the ICLR community more clear, we now revise our Discussion to clearly discuss how the implications from our work can affect the future DNN architectures or activation functions.
> >
> > > Was a capacity-matched control readout tested—for example, a variant applying the same mixing and normalization operations after the ReLU—to isolate the specific contribution of pre-ReLU access to negative features?
> >
> > Although we tested this, because we found little difference in performance, we initially omitted it from Fig. 2a. We now discuss this point in the revised manuscript.
> >
> > > Can the authors clarify whether the psychophysics effects were stronger for neurons with two-tailed response distributions, or whether similar effects occurred for sparse or one-tailed neurons as well?
> >
> > For our psychophysics analysis the participants were given DNN units (10 example units), and V4 model neurons (10 example V4 model neurons)  to estimate the tuning. When we included these two models, we wanted to showcase the difference between two-tailed responses versus one-tailed responses, therefore, our V4 model neurons used in the task are strictly two-tailed and the DNN units used in the task are one-tailed. Since the one-tailed V4 model neurons also exhibit similar unstructured pattern in their anti-preferred images as of the DNN units’ low response images we did not include it. However, we agree that it would be potentially interesting to expand this task to include neurons with different sparsities in both artificial and biological units in the future.

---

> > > ### Author Response · Authors · 2025-11-21
> > > **Cont.**
> > >
> > > > How robust are the findings to the specific image synthesis method used? Would on-manifold synthesis (e.g., diffusion-guided optimization) produce comparable preferred and anti-preferred examples?
> > >
> > > This is a good question. We used a gradient based synthesis approach to generate our model synthesized preferred and anti-preferred images. However, we are aware that there are more sophisticated approaches such as the ones that the Reviewer mentions. One of the main motivations behind developing ImageBeagle was to propose a framework that would be simple, effective, and easy enough for an experimentalist without a computational background to navigate. It would be interesting for future work to compare how the preferred and anti-preferred images selected/generated by these methods differ from each other. We now add a sentence regarding the novel synthesis methods to our revised manuscript.
> > >
> > > > How do the authors view the implications of their results for representation learning in artificial systems? Could introducing bidirectional or opponent-like feature tuning improve generalization or interpretability in DNNs?
> > >
> > > This is an excellent question! We would be excited to see future work that builds anti-preferences into their DNN architectures via better structured activation functions and how this would affect the generalization. Most of the time when DNNs are improved based on biological representations (Dapello et al., ICLR 2023), they result in a more generalizable model, we think this could be the case with anti-preferences. The two-tailed representations double the capacity of a neuron, and due to little to no relationship between anti-preferred and preferred images, they double the amount of information a neuron can have. Therefore, DNNs that have similar tuning properties can represent more information with less neurons, and potentially be more interpretable (e.g. we can view the negative/postive contributions to downstream units).
> > >
> > > > Could the authors comment on how their findings relate to the recent bioRxiv preprint (2025.07.16.665209) reporting suppressive feature selectivity in visual cortex?
> > >
> > > We find 2025.07.16.665209 to be a very relevant work, and we are happy that the quest for anti-preferred images is widespread! It is great that a concurrent work is finding the same thing as in our paper, this does not often happen in science. 2025.07.16.665209 highlights the two-tailedness of neural responses by employing a more extensive experimental analysis. However, they find similar results in that the anti-preferred images that they find (called Least Exciting Images in their work) are structured. We agree with the authors of the paper that feature selective suppression is a crucial characteristic of visual cortical neurons that is often underappreciated. However, we hope that our work along with 2025.07.16.665209 can spur on future neuroscientific experiments that test anti-preferences. Overall, this is a highly relevant paper that we certainly will include in our camera-ready version.

---

> > ### Comment · Reviewer_c1B5 · 2025-11-21
> >
> > Thanks for the detailed feedback, also to the other reviewers' comments. I am happy with those replies and I think my original score is still appropriate.

---

### Official Review · Reviewer_vbPL · 2025-10-31

**Soundness:** 3
**Presentation:** 2
**Contribution:** 3
**Rating:** 4
**Confidence:** 3

**Summary:**

This is a very interesting paper that combines unique experimental data with modern computational techniques to fundamentally make the a potentially groundbreaking observation about neurons in the visual cortex - that is their tuning properties cannot just be characterized by what they prefer (i.e., are excited by), but there is an independent set of stimuli that effectively suppress the activity of a neuron and it is important to identify these anti-preferred stimuli. They show various results to support this conclusion and present a new tool for neuroscientist to use in order to efficiently identify preferred/anti-preferred stimuli. I enjoyed the paper and I think it has important scientific contributions to offer. I don't believe, however, the ICLR venue is the right audience. I know that with a liberal interpretation, one can consider this paper in the scope of the conference. I believe the ICLR paper should either push ML forward or show a new application of an ML method. I think this paper might be considered in the latter category, but the main message is squarely in neuroscience.

**Strengths:**

Solid experimental results.
Very useful and impactful neuroscientific insights.
A public tool for other neuroscientists to use in order to effectively find anti-preferred and preferred stimuli.

**Weaknesses:**

The main weakness in my opinion is that there is little contribution to core or foundational ML. There is also no meaningful innovation on the applied side, where an existing ML tool is applied to a novel problem or a state-of-the-art result is achieved on some standard problem. I also found the style of writing and the format and flow very confusing. It follows the typical general-audience high impact journal format (like Nature), where details of methods and experiments are buried in the back of the paper. To me, a good ICLR paper highlights these up front. In fact, those are supposed to be main contributions.

I found Section 6 to be unconvincing and confusing. I really counldn't follow how the results amounted to the conclusion that "knowing a neuron’s preferred images gives little information about what a neuron’s anti-preferred images will be"

**Questions:**

1) Can you spell out what novel insights you can offer for the ML community?

---

> ### Author Response · Authors · 2025-11-21
>
> We thank the Reviewer for their positive comments of our work and recognizing its contributions. We believe our work fits nicely in the ICLR category of "applications to neuroscience & cognitive science" (https://iclr.cc/Conferences/2026/CallForPapers) and will be of interest to neuroAI researchers that frequently publish in ICLR and related conferences as well as neuroscientists. However, every Reviewer must decide their own threshold, and we respect the Reviewer’s preferences (and anti-preferences!).  Below we address each of the questions and points raised by the Reviewer. If the Reviewer finds our responses and changes to the manuscript sufficient, we request the Reviewer to raise our manuscript’s score, as we are a borderline accept case.
>
> > The main weakness in my opinion is that there is little contribution to core or foundational ML. There is also no meaningful innovation on the applied side, where an existing ML tool is applied to a novel problem or a state-of-the-art result is achieved on some standard problem. I also found the style of writing and the format and flow very confusing. It follows the typical general-audience high impact journal format (like Nature), where details of methods and experiments are buried in the back of the paper. To me, a good ICLR paper highlights these up front. In fact, those are supposed to be main contributions.
>
>
> The emerging field of neuroAI has two goals: 1) identify and use principles of neuroscience to build better AI systems and 2) use AI concepts to better understand the brain. A fundamental concept in ML is the ReLU,  a simple pointwise nonlinearity that, when combined with linear combinations and layers, creates a powerful function approximator. The history of the ReLU, along with other simple pointwise nonlinearities (sigmoid, tanh) is rooted in neuroscience, starting with McCulloch and Pitts. It seems relevant to both the ML community and the neuroAI community (which has a small but growing track at ICLR) to challenge this fundamental notion with real biological data. This will lead to more predictive (Fig. 2a) and biologically realistic neuroAI models of the visual system. There is precedent for ML conference papers to highlight computational neuroscientific findings (e.g., Guo et al., ICML 2022; Dapello et al., ICLR 2023; Geirhos et al., ICLR 2019; Conwell et al., NeurIPS 2021); afterall, NeurIPS starts with “Neural” and ICLR considers “Representations”, presumably how V4 neurons represent stimulus information is relevant. A key goal of papers such as ours is to spur on new experiments, both in neuroscience labs as well as in neuroAI labs focused on improving neuroAI models of the visual system.
> We crafted the manuscript in a logical flow that first identifies two-tailed responses (Fig. 1), validates the existence of anti-preferred stimuli (Fig. 2), confirms that anti-preferred images is a property of a neuron’s representation (Fig. 3), interprets the visual features comprising preferred and anti-preferred images (Fig. 4), and proposes a large-scale tool to identify preferred and anti-preferred natural images (Fig. 5). Given the amount of computational and experimental techniques, including modeling, real experimental data collection, data pruning, human psychophysics, and tool development, we prioritized a narrative that logically flowed versus a Methods→Results, as we simply did not have the room in the main paper. Each analysis helps to establish that anti-preferred images are an important property of V4 stimulus coding.
> We submitted to ICLR because we believe ICLR is the best venue to disseminate the work to diverse fields of core ML researchers, neuroAI researchers, and systems neuroscientists. Our work raises an important, simple question: Does the brain use “ReLUs” differently from DNNs, and if so, why? This gets at the heart of aligning DNN representations to brain representations. Still, the Reviewer may deem our work too much of an outlier for the ICLR community; we respect this opinion, but if this is the majority opinion, it sets up a large barrier for the neuroAI community to share results with the broader ML community.
>
> We address further points below.

---

> > ### Author Response · Authors · 2025-11-21
> > **Cont.**
> >
> > > I found Section 6 to be unconvincing and confusing. I really counldn't follow how the results amounted to the conclusion that "knowing a neuron’s preferred images gives little information about what a neuron’s anti-preferred images will be"
> >
> > We apologize for the confusion. The goal for Section 6 is to see if we can interpret the visual features of the anti-preferred images, and how these features relate to preferred features. We now plan to revise Section 6 to include a more extensive analysis on the comparison between preferred and anti-preferred images. Specifically, we will add a new figure that will extend our findings from Fig. 4e to clarify that we do not find a relationship between anti-preferred images and preferred images across the V4 model neurons. In Fig. 4e we show this via interpretable features of the top 100 preferred and top 100 anti-preferred images of a given V4 model neuron, where we plot the difference between the preferred and anti-preferred values for a given image statistic (contrast, luminance, orientation, etc.) for every V4 model neuron. As we discuss it in the manuscript, from this we find that across the population there is not a clear similarity between preferred and anti-preferred images. In our new figure, we will display the similarity matrix between the preferred and anti-preferred images across all neurons via using both interpretable features and DNN features. In addition, we will include a two-step KNN analysis where we plot the clustering within and across the preferred and anti-preferred features. Our results indicate that there is little to no relationship between preferred and anti-preferred images, in other words, one can sample two features, make one preferred and one anti-preferred, and staple them together, forming a V4 neuron’s tuning. This suggests that a V4 population effectively doubles its feature selectivity, as downstream neurons can extract both preferred and anti-preferred features. Our second new analysis confirms this with a downstream object recognition task: A population whose neurons have two-tailed response distributions achieve the same prediction with half as many neurons as a population with neurons that only have one-tailed response distributions.
> >
> > > Can you spell out what novel insights you can offer for the ML community?
> >
> > - We offer a new mapping from features to neural responses that leverage the anti-preferences. This might be a small but potentially important change, considering BrainScore benchmark only uses PCA linear regression.
> > - We offer important insights into neural coding; how current DNNs differ from biological neurons may offer new insights into improving DNN performance.
> > - We build on previous work in neuroAI (Guo et al., ICML 2022, Dapello et al., ICLR 2023) to push towards biologically inspired DNN architectures for more brain-like object recognition.
> > - With our work we propose; 1) a general principle of stimulus encoding in the visual cortex, 2) neuroscientific experiments to confirm anti-preferred images, and 3) computational modeling and human psychophysics to reveal the significance of anti-preferred images.
> >
> > We understand that the Reviewer’s primary criticism was that our paper was descriptive rather than offering an algorithmic/architecture advance. Descriptive papers do have a place at ICLR, the descriptive papers of adversarial examples and shape/texture biases in DNNs have opened entirely new research directions. Our work is particularly timely for neuroAI researchers eager to transform neuroscience principles into new model architectures. Beyond the rigorous testing of our anti-preferred image hypothesis, we also present a new tool (ImageBeagle) to efficiently identify anti-preferred natural images, this tool will be of use both to neuroscientists and interpretable ML researchers eager to understand the inner workings of DNNs. Our results provide new insights about neural tuning, catalyzing new closed-loop experiments to optimize anti-preferred images for other brain areas and a new way to compare DNNs and real neurons.
> > To make our contributions clearer, we now emphasize them in the Discussion.

---

### Official Review · Reviewer_qR7N · 2025-10-31

**Soundness:** 3
**Presentation:** 3
**Contribution:** 2
**Rating:** 2
**Confidence:** 3

**Summary:**

This paper overturns the traditional “one-tail” view of visual neuron selectivity by showing that many macaque V4 neurons exhibit two-tailed responses, they react strongly to both preferred and anti-preferred stimuli. These anti-preferred images are rich and structured, not blank, and are essential for accurately modeling neural tuning. Incorporating both stimulus types improves encoding performance and aligns with human perception, unlike ReLU-based DNNs that show only one-tailed selectivity. The authors also introduce ImageBeagle, a scalable tool for discovering preferred and anti-preferred stimuli across millions of natural images.

**Strengths:**

* This work provides an interesting analysis of the preferred and anti-preferred stimuli from macaque V4 region
* The modeling and analysis are robust. Author use multiple experiments: encoding models, data pruning experiments, and psychophysics tasks to show that both preferred and anti-preferred stimuli are essential to fully capture a neuron’s tuning curve.
* The author released ImageBeagle, an efficient large-scale natural image search tool. Provides practical value to visual neuroscientists interested in optimizing neurons’ responses

**Weaknesses:**

* The discovery of “two-tailed” response distributions is presented as novel, but prior work has long recognized that neurons exhibit both excitatory (maximizing) and suppressive (inhibitory) responses to different stimuli. The paper does not clearly distinguish how its findings go beyond this established understanding.
* The comparison with linear-nonlinear (LN) models and task-driven DNNs is somewhat misleading. Their one-tailed response patterns largely reflect architectural constraints (e.g., ReLU activation), not an intrinsic inability to model two-tailed selectivity. More recent encoding models [1] can predict both positive and negative response values, so the contrast with “biological two-tailed neurons” is overstated.
* The neuron tuning analysis relies on a single model type and uses only the R² metric to assess performance. This is insufficient to robustly support the claim that including anti-preferred stimuli improves tuning prediction. Additional analyses on more encoding models and metricstrained with the same set of data would strengthen the evidence.
* Although the paper analyzes anti-preferred images from several perspectives—neural recordings, modeling, and psychophysics—the overall impact of these analyses is limited. Beyond showing that incorporating both preferred and anti-preferred images can improve encoding model performance in low-data regimes, the study provides little insight into how anti-preferences fundamentally contribute to neural computation or representation.

[1] Aria Y Wang, Kendrick Kay, Thomas Naselaris, Michael J Tarr, and Leila Wehbe. Better models of human high-level visual cortex emerge from natural language supervision with a large and diverse dataset. Nature Machine Intelligence, 5 (12):1415–1426, 2023. 2

**Questions:**

* The improved tuning performance using both preferred and anti-preferred images may mainly result from greater response diversity, instead of the features of the images. Random images yield weaker predictions likely because their responses are less extreme, but outperform when more data are available. The claimed connection between this result and the neuron’s tuning, specifically that preferred and anti-preferred images reveal identifiable visual features learned by the model requires further verification.
* Could the authors clarify what is being predicted in this analysis? Are they training a model to predict the activation responses of individual ResNet-50 units to images? If so, is this model evaluated with actual measured biological responses to validate this comparison?
* The ImageBeagle algorithm searches through nearest neighbors in image feature space without incorporating neural objectives. How should we interpret the “response curve” produced during this search as reflecting neural response changes? Is the method assuming that smooth interpolation between nearby images in feature space implies a correspondingly smooth interpolation of neural responses? If so, could the authors provide justification or evidence supporting this assumption?

---

> ### Author Response · Authors · 2025-11-21
>
> We thank the reviewer for highlighting the robustness of our results and contribution to the community. We are surprised by the Reviewer’s score of 2, typically reserved for unpolished or irrelevant work. We believe our work is polished, with diverse computational and experimental analyses, and of large interest in ICLR’s track for ‘applications to neuroscience & cognitive science’. Our work challenges the assumption of ReLU activation functions commonly used in deep neural network models of visual cortex; this is highly relevant to the emerging neuroAI field.  Below we address the questions and concerns raised by the Reviewer. We ask the Reviewer, if satisfied with our responses and planned revisions, to consider increasing the manuscript’s score, as our paper is a borderline accept.
>
> > The discovery of “two-tailed” response distributions is presented as novel, but prior work has long recognized that neurons exhibit both excitatory (maximizing) and suppressive (inhibitory) responses to different stimuli. The paper does not clearly distinguish how its findings go beyond this established understanding.
>
> We want to clarify that we do not claim that our work is the first to propose that neurons in the visual cortex exhibit both excitatory and inhibitory response; we cite prior work in our manuscript. For example, in early visual areas like V1, it’s known that the anti-preferred orientation of a V1 neuron is usually orthogonal to that of its preferred orientation, see L075-L078 in the manuscript; it remains unclear what a V1 neuron’s anti-preferred image is when considering all possible natural images. Moreover, to our knowledge there has not been any studies identifying the anti-preferred images of V4 neurons (see Franke et al., 2025 for recent concurrent, complementary work). Most work that look into the feature tuning in visual cortex to natural images (Földiák, 2001; Benda et al., 2007; Yamane et al., 2008; Okazawa et al., 2015; Cowley et al., 2017a; Cadieu et al., 2007; Berardino et al., 2017; Abbasi-Asl et al., 2018; Pospisil et al., 2018; Ponce et al., 2019; Bashivan et al., 2019; Walker et al., 2019; Gu et al., 2022; Cowley et al., 2023; Willeke et al., 2023; Pierzchlewicz et al., 2024; Wang and Ponce, 2024) specifically focus on the preferred images (also known as maximally exciting images), and do not discuss or highlight the role of anti-preferred images.
>  As we state in L075, the existence of anti-preferred images in higher-order visual cortical areas is not obvious since the feature selectivity is more complex. In fact, the feature selectivity of higher-order visual cortex was one of the key motivators of using nonlinear activation functions, like ReLU (Glorot et al., 2011; Fukushima 1980).
>
> We address further points below.

---

> > ### Author Response · Authors · 2025-11-21
> > **Cont.**
> >
> > > The comparison with linear-nonlinear (LN) models and task-driven DNNs is somewhat misleading. Their one-tailed response patterns largely reflect architectural constraints (e.g., ReLU activation), not an intrinsic inability to model two-tailed selectivity. More recent encoding models [1] can predict both positive and negative response values, so the contrast with “biological two-tailed neurons” is overstated.
> >
> > We discuss LN models briefly in the introduction to draw the similarities between the feature selectivity in statistical models inspired by the quest for identifying the preferred images of visual cortical neurons. There is a very rich literature (Ponce et al., 2019; Bashivan et al., 2019; Walker et al., 2019; Gu et al., 2022; Cowley et al., 2023; Willeke et al., 2023; Pierzchlewicz et al., 2024) that seeks to identify the preferred images for understanding the neural tuning in the visual cortex. In fact we discuss how these architectural constraints for both LN models and task-driven models were inspired by the selectivity patterns similar to selectivity found in the visual cortex.
> > Furthermore, we are confused about the Reviewers remarks “Their one-tailed response patterns largely reflect architectural constraints (e.g., ReLU activation), not an intrinsic ability to model two-tailed selectivity”. A goal in neuroAI is to build biologically-realistic models of brain processing; that DNN units are one-tailed is biologically unrealistic, given our observations of two-tailed V4 responses. ReLU activations were influenced by the thresholding in the brain (e.g., neurons can only have positive spike counts), i.e., this architecture decision was inspired by the brain’s physiology. That pre-ReLU activations perform worse at predicting V4 responses than post-ReLU activations (Fig. 2a) further highlights the mismatch between DNN units and neurons in biological realism. Thus, even though biological architecture suggests that neurons should implement ReLU like functions, we find that V4 neurons do not follow this behavior. This divergence between the ReLU activations and V4 responses was not previously proposed, yet we believe it can have a strong impact in the way we think about the role and design of activation functions in DNNs.
> > In addition, we do not claim that encoding models are unable to predict two-tailed responses (in fact, the ReLUs allow for better prediction by splicing together one-tail responses). However, our encoding models should reflect the underlying biology, and individual DNN units should realistically reflect individual V4 neurons. This is not the case for most DNN models. Moreover, the paper that the Reviewer cites is for fMRI voxels where each voxel is a sum of 100,000s of neurons, making two-tails likely via central limit theorem (e.g., creating Gaussian-like response distributions). Beyond this, the paper does not perform experimental validation to verify that the predicted anti-preferred images really do suppress the voxel activity. Here, we perform closed-loop neuroscientific experiments to confirm that our models identify the anti-preferred images (Fig. 2b) which is a very strong and robust method of validation. Thus, to suggest that [1] refutes our work is a misinterpretation of the literature.
> >
> > > The neuron tuning analysis relies on a single model type and uses only the R² metric to assess performance. This is insufficient to robustly support the claim that including anti-preferred stimuli improves tuning prediction. Additional analyses on more encoding models and metrics trained with the same set of data would strengthen the evidence.
> >
> > We performed the analyses in Figure 3b with the most predictive models of V4 neurons in the field (Cowley et al., 2023), which achieve noise-corrected R²=0.6 versus other task-driven DNN models at noise-corrected R² at 45%. Noise-corrected R² is the standard in the field (see BrainScore website, Schrimpf et al., 2020). We note this is a version of Pearon’s correlation squared, not coefficient of determination. We are unaware of other metrics useful in this comparison and encourage the Reviewer to list such metrics. We validated these models with closed-loop validation experiments (Fig. 2b), indicating these models are able to identify the preferred and anti-preferred features of these V4 neurons. We are currently analyzing the predicted responses of AlexNet (along with its DNN units) to further strengthen and generalize the results in Figure 3.

---

> > > ### Author Response · Authors · 2025-11-21
> > > **Cont.**
> > >
> > > > Although the paper analyzes anti-preferred images from several perspectives—neural recordings, modeling, and psychophysics—the overall impact of these analyses is limited. Beyond showing that incorporating both preferred and anti-preferred images can improve encoding model performance in low-data regimes, the study provides little insight into how anti-preferences fundamentally contribute to neural computation or representation.
> > >
> > > The existence of anti-preferred images itself fundamentally changes how we view the neural computations of V4 neurons. This finding, in our opinion, is significant enough to pursue with neural recordings, closed-loop experiments, modeling simulations, and human psychophysics, to leave no doubts of their existence. Why V4 neurons have anti-preferred images is an intriguing question that will require decades of experiments and modeling to understand. We propose an initial hypothesis that anti-preferred images allow a V4 population to effectively double its feature selectivity. In other words, the V4 population can transmit double the amount of information; we will include a new analysis that tests this hypothesis by decoding the V4 model responses in an object recognition task.  Understanding how anti-preferences contribute to downstream processing and ultimately to behavior in a single paper is too high a bar, considering the field itself has little insight into how preferred features of V4 neurons fundamentally contribute to neural computation or representation. Our work follows a long line of neuroAI research in machine learning conferences that highlight differences between object recognition DNNs and visual cortex in an effort to better align the two (Dapello et al., NeurIPS 2020;  Safarani et al., NeurIPS 2021; Guo et al., ICML 2022; Linsley et al., NeurIPS 2023; Dapello et al., ICLR 2023). Answering these questions will require decades of work with neural recordings, neural perturbations, and anatomical tracing.
> > >
> > > > The improved tuning performance using both preferred and anti-preferred images may mainly result from greater response diversity, instead of the features of the images. Random images yield weaker predictions likely because their responses are less extreme, but outperform when more data are available. The claimed connection between this result and the neuron’s tuning, specifically that preferred and anti-preferred images reveal identifiable visual features learned by the model requires further verification.
> > >
> > > We believe our experimental validation does support the existence of anti-preferred images and show that they are highly structured (see Fig. 2b). The same experimental validation framework has been employed in many recent studies to identify the preferred images of V4 neurons (Bashivan et al., 2019; Cowley et al., 2023, Pierzchlewicz et al., 2023; Willeke et al., 2023), but do not consider anti-preferred images.  The Reviewer’s point about our data pruning analysis (Fig. 3a-b) is confusing in light of the same analysis performed on DNN units (Fig. 3c), which indicates that preferred images alone (which are likely more extreme than the weaker responses to randomly-chosen images) outperform randomly-chosen images in this setting. In addition, the analysis in Figure 3 is to verify that both preferred and anti-preferred images are enough to capture a V4 neuron’s tuning (with the obvious caveat that with enough data, randomly-choosing images trumps most data pruning strategies). This analysis indicates that anti-preferred images contribute to a V4 neuron’s tuning. This motivated us to then investigate the “identifiable visual features”, which we explore in Figure 4, where we compute the image statistics separately for preferred and anti-preferred images (see Fig. 4d-e). In our work, no single analysis captures the importance of anti-preferred images; thus, we rely on a suite of techniques, experiments, and analyses.
> > >
> > > > Could the authors clarify what is being predicted in this analysis? Are they training a model to predict the activation responses of individual ResNet-50 units to images? If so, is this model evaluated with actual measured biological responses to validate this comparison?
> > >
> > > In Fig. 3, we measure the information content of anti-preferred images by including them or leaving them out when estimating a neuron’s tuning. We do this for 219 V4 model neurons (exhibit similar two-tailed response distributions like real V4 neurons) (Fig. 3b) and 219 randomly selected DNN units (Fig. 3c). For each trace, we train the models with the selected set of images and evaluate their predictive performance to the same randomly selected held-out images. For more details,  please see A.1.3 in the appendix. Importantly, we chose these encoding models because they accurately predict the preferred and anti-preferred features of V4 neurons via closed-loop experiments. This experimental validation is beyond what is expected and found in typical modeling papers.

---

> > > > ### Author Response · Authors · 2025-11-21
> > > > **Cont.**
> > > >
> > > > > The ImageBeagle algorithm searches through nearest neighbors in image feature space without incorporating neural objectives. How should we interpret the “response curve” produced during this search as reflecting neural response changes? Is the method assuming that smooth interpolation between nearby images in feature space implies a correspondingly smooth interpolation of neural responses? If so, could the authors provide justification or evidence supporting this assumption?
> > > >
> > > > One goal of our paper is to inspire future experiments to test for the anti-preferred images of visual cortical neurons. This requires neuroscientists to identify a neuron’s anti-preferred images with as little recording time as possible. Current approaches typically use gradient techniques to synthesize preferred images; one drawback of this approach is that the experimenters need technical skill in using DNN models as well as access to dedicated GPUs, this is typically not the case for the standard visual neuroscience laboratory. We designed ImageBeagle to fill this gap, it is easy to understand how it works, and no GPUs are necessary to run it (unless the user is running a model).
> > > >
> > > > We tested ImageBeagle on our encoding models of V4 neurons; this is how we identified a “natural-image” tuning curve in Fig. 5d. We included this analysis to motivate future experiments to try this approach; we have not tested it with real neural data. The assumption the Reviewer brings up is an interesting one, are neural responses smooth to smoothly-varying parameters of natural images (Wang and Ponce, arXiv, 2022, Cowley et al., 2023)? This question can be better answered with ImageBeagle, which has identified the nearest neighbors of millions of images. Regardless, the main issue that ImageBeagle solves in this setting is to ensure the images themselves smoothly vary, this is often not the case when presenting randomly-chosen images, sorting the responses, and displaying the tuning curve. In this sense, ImageBeagle identifies a single parameter (that varies nonlinearly in image space) for which the response is guaranteed to vary over between its preferred and anti-preferred images.

---

### Official Review · Reviewer_iGeu · 2025-11-01

**Soundness:** 2
**Presentation:** 3
**Contribution:** 4
**Rating:** 6
**Confidence:** 4

**Summary:**

A large body of work exists on studying the properties of stimuli that strongly (or even 'maximally') excite a neuron in visual cortex. The manuscript at hand extends this approach by exploring anti-preferred stimuli that lead to very weak neural responses. The tuning properties of V4 neurons, as well as those of other areas are shown to exhibit a two-tailed distribution, showing that both highly preferred as well as anti-preferred stimuli exist. This is not mirrored by artificial neural networks. The authors derive an alternative mapping from DNN units to real neurons which yields better predictive performance. It is shown that models perform best when both preferred and anti-preferred stimuli are included during training. The authors describe the results of a psychophysics experiment showing that humans can better predict neural responses to novel stimuli when given both preferred and anti-preferred stimuli as a reference. Finally, the authors propose a method to more efficiently sample images to determine the strongest and weakest activators, yielding large computational savings.

**Strengths:**

1. Studying anti-preferred stimuli is an important step towards capturing neural tuning properties more globally, beyond exceptionally strong activators.
2. The results of the neural data analysis convincingly demonstrate the two-tailed response distribution.
3. The discussion of ReLU-based DNNs in this context is interesting and shows a clear computational difference between artificial neural networks and the primate visual system.
4. The psychophysics experiment yields interesting insights, showing that anti-preferred stimuli have additional merit for interpreting a neuron's tuning.
5. ImageBeagle demonstrates substantial benefits over random image sampling. The motivation for the method, as well as the discussion of synthetic MEIs is well-rounded and it is made clear to the reader how ImageBeagle combines the advantages of natural-image sampling with computational efficiency of MEIs.
6. The work is well presented; Figures are clear and intuitive and it is easy to understand the results. The work is well motivated and the main story is clear.

**Weaknesses:**

I am not fully convinced by the results in sections 3) and 4). I would raise the following points:

1.
1.1 The existence of anti-preferred stimuli suggests that neural responses do not follow a ReLU-like response pattern. Based on this fact, the work argues that a linear map from ReLU-thresholded DNNs units (standard in the field) should be suboptimal. I find this hypothesis to not be well motivated in the text. The manuscript clearly states that anti-preferred images contain some anti-preferred feature which suppresses responses. As the post-ReLU units act as one-tailed detectors for preferred features (demonstrated nicely here!), I see no reason why a map like this should be suboptimal. If a feature is anti-preferred, a ReLU unit detecting that features could simply be assigned a negative weight in the linear map. I would ask the authors to clarify the motivation for this section.
1.2 Further, the experiment conducted to prove the hypothesis does not seem that convincing to me. Mixing the features linearly, applying a ReLU and then a final linear map increases the capacity of the readout model, which could explain better performance without any real ties to the presented hypothesis. The fact that the control experiment with removed intermediate ReLUs does not increase performance does not seem surprising as that procedure again yields a linear map. I therefore find the claim in line 198f to be very strong given the data.

2. In Section 4, it seems to me like the experiment is biased towards finding pref+anti-pref to be the best condition. If only (anti-)preferred images are shown, we would expect the average response on the training data to be higher/lower than on the test data, which would artificially reduce R^2. Therefore, it would be expected that 'random' outperforms either of the two biased training sets and I could imagine that the superiority of 'pref+anti-pref' is a methodological artifact. I would strongly suggest that the authors reproduce the results with a mean-independent statistic like correlation between predicted and true responses.

Some further points:

3. Much of the analysis (e.g. Section 5) is based on digital twins rather than real neural data due to noisiness. While this limits the conclusions one can draw a little bit, I realize that this is a difficult issue to address and do not see it as a big problem.
4. ImageBeagle, as a strong contribution of the paper, is not explained thoroughly enough in the main text, in my opinion. I agree it makes sense to put small details in the appendix, but in the current form it is difficult to understand from the main text alone. I would kindly ask the authors to include some more detail in the main text.
5. For ImageBeage specifically, it is unclear how strongly the computational savings are overestimated by evaluating on the digital twin model rather than real neural responses.

Overall, I do like this work and the contributions are strong. However, I would ask the authors to consider the points I've raised regarding some of the methodology. If these are addressed, I would increase my score and argue the paper should be accepted.

**Questions:**

How does ImageBeagle transfer across brain regions? Does the graph need to be recomputed for different regions as the similarity should be computed at different DNN layers?

**Details Of Ethics Concerns:**

The work contains animal experiments.

---

> ### Author Response · Authors · 2025-11-21
>
> We thank the Reviewer for praising the presentation and motivation of our work.  Below we address each of the questions and points raised by the Reviewer.
>
> > 1.1
>
> This is an excellent point that we need to better explain. The pre-ReLU and post-ReLU units both comprise ~1k filters with different feature selectivities. The post-ReLU units provide ~1k “one-tailed” features which we can combine (with either positive or negative weights) to craft the predicted tuning of the V4 responses. The pre-ReLU units provide ~1k “two-tailed” features…we can access either the preferred or anti-preferred features of these pre-ReLU units by including a ReLU layer in our mapping (Fig. 2a, ‘v’). These 1k pre-ReLU “two-tailed” units can then be thought of as 2k “one-tailed” units (a doubling of feature selectivity). This allows the model access to double the number of features without the need to explicitly have ~2k filters. Your comment motivated us to understand this better in a downstream task of object recognition; indeed, we find that the two-tailed pre-ReLU units achieve a similar accuracy to a population of 2x post-ReLU units. This suggests that one reason V4 neurons have both preferred and anti-preferred features, and that these features are largely independent of one another (Fig. 3 and new clustering analysis), is to effectively double its feature selectivity.
>
> > 1.2
>
> First, we should make clear that the analysis in Fig. 2a does not conclusively prove the existence of anti-preferred images. Our closed-loop validation experiment (Fig. 2b) provides substantially stronger evidence of this. Our goal is to build up a suite of analyses to convince ourselves that anti-preferred images are a meaningful property of V4 tuning. The purpose of the Fig. 2a analysis is to highlight that assuming that V4 neurons do have anti-preferred images, we should expect that a linear-relu-linear model outperforms other types of simple mappings. Our Fig. 2a results also answer a puzzling question in the field of why pre-ReLU activity performs worse at predicting V4 responses than post-ReLU activity. To our knowledge, the benefits of a simple linear-relu-linear model for predicting V4 responses has not been proposed or demonstrated in the field (e.g., BrainScore benchmark only uses linear mappings). Taken together, we believe our claim in 198 “This algorithmic improvement is a direct result of assuming that V4 neurons encode anti-preferred images.” is correct, in the sense that we hypothesized the linear-relu-linear model would work based on the assumption of anti-preferred images.
> X→Y does not imply Y→X; as the Reviewer mentions, observing Y (e.g., that linear-relu-linear performs better) does not necessarily make X true (e.g., it may simply be due to higher capacity). For example, if the V4 neurons were truly one-tailed, it still may be the case that the linear-relu-linear model outperforms the other tested models. While we could further investigate this in this analysis, we believe our other analyses, experiments, and psychophysics better proves the importance of anti-preferred images. Instead, Fig. 2a serves as a demonstration that realizing the existence of anti-preferred images leads to the practical result of better estimation of V4 tuning.
>
> > 2
>
> We apologize, our $R^2$ metric is Pearson correlation squared, which accounts for differences in mean, scale, or sign between predicted and true responses; we do not use coefficient of determination, which is affected by overall means. Thus, the effect the Reviewer mentions cannot explain our results. We now clarify this in Section 4.
>
> > 3
>
> The reviewer is correct that indeed it is hard to perform these analyses based on real neural data due to the inherent noise in neural responses. Since V4 responses are noisy (Poisson-like); this noise hinders the subject’s ability to infer the underlying tuning. Moreover, there is also the data limitation problem if one wanted to do this analysis with real V4 responses. With *in silico* models, we can choose pairs of stimuli from 500k candidate images, whereas a typical recording session presents ~1k images, as recording time is limited. 1k images are likely not enough to fully describe a neuron’s tuning, especially for the preferred and anti-preferred images. Importantly, we used the most predictive models currently available in the field, and confirmed via closed-loop experiments that the anti-preferred images predicted by these models suppressed responses of real V4 neurons.
> Our motivation for this entire work is to inspire new experiments to identify anti-preferred images in V1, V4, and IT. A key part of this motivation is to ensure our results hold in model simulations. We believe we now have enough evidence via these simulations to persuade neuroscientists to collect more data to test the hypotheses proposed in this work.
>
> We address further points below.

---

> > ### Author Response · Authors · 2025-11-21
> > **Cont.**
> >
> > > 4
> >
> > We thank the Reviewer for their suggestion. Since ImageBeagle is more of an application we developed for future research, and due to limited space, we cut down the text substantially in that section, leaving a lot of details in Methods and Supplemental. Given the extra page in the camera-ready version, we will better motivate and explain the ImageBeagle section.
> >
> > > 5
> >
> > We now add this as a limitation of our work; this requires further closed-loop experiments to assess the effectiveness of ImageBeagle. We leave this for future work,  given the diversity of approaches to obtain preferred and anti-preferred images in the field, the time has come for a definitive study to place methods (both image-choosing and image-synthesizing) head-to-head to see which ones are most effective. Regardless, we believe the simplicity of ImageBeagle will be attractive to many visual neuroscience labs that do not have access to the technical expertise needed to synthesize preferred and anti-preferred images.
> >
> > > How does ImageBeagle transfer across brain regions? Does the graph need to be recomputed for different regions as the similarity should be computed at different DNN layers?
> >
> > We find that ImageBeagle distance estimation is enough for almost all of the visual tasks needed by visual neuroscientists, as these embeddings are enough to find human perceptually similar nearest neighbors (for example nearest neighbors for randomly-chosen images in the Appendix, see Supplementary Figure 4).  We will also include the results for V1 and IT neurons to supplement our V4 results and include them in our camera-ready version. Based on preliminary results, ImageBeagle’s performance remains the same.
> >
> > We hope that our responses and planned revisions have sufficiently addressed all concerns and questions raised by the Reviewer. If this is the case, we hope the Reviewer considers updating their score accordingly, as our paper is a borderline-accept case. If there are still any questions or concerns, we would be happy to clarify and engage in further discussion.

---

> ### Comment · Reviewer_iGeu · 2025-11-24
>
> Thank you for the interesting elaboration; I found most of the answers convincing. However, in my opinion the weaknesses raised in 1. are not fully addressed:
>
> 1.1 I do understand the motivation you state, however it seems to hinge on the fact that the model neurons are themselves 'two-tailed' feature extractors, which appears to be a clear contradiction to the results presented in 118ff.
>
> 1.2. Perhaps I was being a little ambiguous in my original comment, when saying 'Further, the experiment conducted to prove the hypothesis does not seem that convincing to me'. The hypothesis I was referring to was not the overall hypothesis of the paper, but only the hypothesis that the standard post-ReLU readout is suboptimal due to the two-tailedness of response distributions. I am not debating the novelty of the linear-ReLU-linear model but, given enough data, it is not surprising that higher-capacity models perform better. The connection of this finding to the two-tailedness seems weak to me. In my opinion you are hitting the nail on the head in your response:
>
> > observing Y (e.g., that linear-relu-linear performs better) does not necessarily make X true (e.g., it may simply be due to higher capacity). For example, if the V4 neurons were truly one-tailed, it still may be the case that the linear-relu-linear model outperforms the other tested models. While we could further investigate this in this analysis, we believe our other analyses, experiments, and psychophysics better proves the importance of anti-preferred images.
>
> While I agree that the other analysis _do_ highlight the importance of anti-preferred images, I think this particular experiment does not. Therefore, I maintain my position that the claim “This algorithmic improvement is a direct result of assuming that V4 neurons encode anti-preferred images." is not really justified based on the presented results.
>
>
> For the sake of brevity, I have only focused on the negative points in this response. Overall, I think this is a good paper and the remaining issues I see are minor enough to be evaluated and considered by the reader. I think the work is suitable for publication at ICLR.
>
> EDIT: To reflect this, I have increased my score to 8.

---

### Official Review · Reviewer_Bncs · 2025-11-10

**Soundness:** 3
**Presentation:** 3
**Contribution:** 3
**Rating:** 6
**Confidence:** 4

**Summary:**

This paper revisits the classic notion of neuronal selectivity by demonstrating that neurons in the primate visual cortex, particularly area V4, exhibit two-tailed response distributions: responding not only to preferred stimuli that excite activity but also to distinct anti-preferred stimuli that suppress it. Using electrophysiological recordings, modeling, and psychophysical experiments, the authors show that these anti-preferred images are structured and informative, not merely blank or featureless, and that incorporating them significantly improves decoding and predictive models of neural responses. Human participants also rely on both preferred and anti-preferred examples to infer neural tuning, whereas deep neural networks, constrained by ReLU activations, typically capture only one-tailed preferences. The study further introduces ImageBeagle, a tool for efficiently identifying preferred and anti-preferred stimuli across millions of natural images. Overall, the paper establishes anti-preferred stimuli as a key component of neural encoding in visual cortex, expanding the representational capacity of neurons and suggesting new directions for biologically inspired model design.

**Strengths:**

The manuscript is easy to understand and well-organized. It provides  intuitive figures and explanations that make complex concepts accessible to both neuroscience and machine learning audiences.The paper presents a good amount  of well-controlled experimental results combining neural recordings, computational modeling, and psychophysics, which together provide strong empirical support for the existence and importance of anti-preferred stimuli in visual encoding. The authors also contribute a valuable open-source tool, ImageBeagle, which enables researchers to efficiently identify preferred and anti-preferred stimuli across large natural image datasets, potentially accelerating future neuroscience studies. In addition,

**Weaknesses:**

Although the paper provides strong evidence for two-tailed neural responses, it mainly examines a single model based on ResNet-50 features and their mapping to V4 activity. It remains unclear whether the same results would generalize to other architectures or visual areas. The finding that combining high- and low-activation images improves decoding is somewhat expected, since training with a broader range of responses typically enhances prediction accuracy. What is more surprising is that anti-preferred stimuli alone do not provide more predictive power than random images (Fig. 3), suggesting that the information they convey is only useful when combined with preferred responses. Finally, while the experimental work is extensive, several analyses rely heavily on model-based predictions rather than fully independent data, which somewhat limits the strength of the causal conclusions.

**Questions:**

Your analysis focuses on mappings from ResNet-50 features to V4 activity. Have you tested whether similar two-tailed response patterns appear when using other architectures (e.g., self-supervised or non-ReLU models) or in other visual areas such as IT or V1?

In Figure 3, training on anti-preferred stimuli alone performs about as well as random sampling. How should we interpret this in light of your claim that anti-preferred stimuli are an important part of neural encoding? Do they carry independent information, or mainly complementary information when combined with preferred images?

Could you elaborate on the possible biological mechanism that gives rise to these two-tailed response distributions? For example, do you suspect inhibitory circuitry or normalization processes in V4 as the source of the anti-preferred suppression?

You show that combining pre-ReLU features improves prediction. Have you tried using activation functions other than ReLU (e.g., tanh or leaky-ReLU) to see whether two-tailed behavior naturally emerges in artificial networks?

The ImageBeagle tool seems promising. Can you clarify how well it generalizes across different neurons or datasets? For instance, would it work for V1 or IT recordings, or for neurons modeled with non-DNN representations?

In the human experiments, participants performed better when shown both preferred and anti-preferred images. Do you think this reflects an actual perceptual analogy to how V4 represents visual information, or is it more an artifact of the task design?

---

> ### Author Response · Authors · 2025-11-20
>
> We thank the Reviewer for finding our work relevant for neuroscience and machine learning communities and praising the presentation of our manuscript.  Below we address each of the questions and points raised by the Reviewer. If the Reviewer finds our responses and changes to the manuscript sufficient, we request the Reviewer to raise our manuscript’s score, as we are a borderline accept case.
>
> > Although the paper provides strong evidence for two-tailed neural responses, it mainly examines a single model based on ResNet-50 features and their mapping to V4 activity.
>
> Our choice on using ResNet-50 as our surrogate DNN model comes from the vision neuroscience literature where mid-layer ResNet-50 features have been shown to be one of the most predictive task-driven DNN models of V4 neural responses (Cowley et al., 2023; Cadena et al., 2024), compared to other task-driven units. However, our findings are not specific to ResNet-50, we also evaluated the skewness of 4 other popular DNN models (see A.3 in the appendix) and found similar skewness values to that of ResNet-50. To make this generalizability more concrete, we are currently performing the same analyses for another predictive model, AlexNet, and we will include the results in the final paper.
>
> > It remains unclear whether the same results would generalize to other architectures or visual areas.
>
> We assess the skewness of other visual cortical areas, V1 and IT, in Fig.1d alongside V4. Our results from this allude that other cortical areas in the visual hierarchy also exhibit different degrees of two-tailedness. For the rest of the paper we focus our analyses in V4 due to the availability of a large dataset and the availability of highly predictive models for predicting the neural responses.
>
> > The finding that combining high- and low-activation images improves decoding is somewhat expected, since training with a broader range of responses typically enhances prediction accuracy. [...]
>
> Both preferred images and anti-preferred images alone fail to outperform random images for the V4 neuron models (Fig. 3b). For DNN units, that was not the case, the preferred images outperformed randomly-chosen images (Fig. 3c). This suggests that both  are needed to capture a V4 neuron’s tuning (i.e. both preferred and anti-preferred images are informative of a V4 neuron’s tuning). This does not imply that anti-preferred images alone are not useful, downstream neurons have access to these features to perform visual tasks, such as object recognition. We will include a new analysis that shows this on our camera-ready version if we don’t have enough time to upload the revised the manuscript before December 3rd.
>
> > Finally, while the experimental work is extensive, several analyses rely heavily on model-based predictions rather than fully independent data, which somewhat limits the strength of the causal conclusions.
>
> We rely on model simulations because current recording technology does not allow the collection of responses to many images (greater than 1,000), and experimental data is expensive to collect. One goal of our work is to convince neuroscientists to collect responses to anti-preferred images to better investigate the ideas in this work, e.g., let theory guide experiment. In addition, while model-based predictions might not replace the true V4 responses, we believe our choice of using an experimentally validated surrogate model further supports the validity of our results.
>
> Below we address the **Questions** in a new comment.

---

> > ### Author Response · Authors · 2025-11-21
> > **Questions**
> >
> > > Your analysis focuses on mappings from ResNet-50 features to V4 activity. Have you tested whether similar two-tailed response patterns appear when using other architectures (e.g., self-supervised or non-ReLU models) or in other visual areas such as IT or V1?
> >
> > In Fig.1d, we show the skewness of the real neural responses from V1, V4, and IT as well as equivalent DNN units from ResNet-50. In addition, we compute the skewness for 4 other DNN models (Supplementary Figure 6) and find that most of them exhibit a behavior similar to ResNet-50.
> >
> > > In Figure 3, training on anti-preferred stimuli alone performs about as well as random sampling. How should we interpret this in light of your claim that anti-preferred stimuli are an important part of neural encoding? Do they carry independent information, or mainly complementary information when combined with preferred images?
> >
> > The goal of Figure 3 is to show that a neuron’s tuning is more than its preferred images (the current notion in the field, inspiring many studies to identify 'maximally exciting images', see refs (Földiák, 2001; Benda et al., 2007; Yamane et al., 2008; Okazawa et al., 2015; Cowley et al., 2017; Cadieu et al., 2007; Berardino et al., 2017; Abbasi-Asl et al., 2018; Pospisil et al., 2018; Ponce et al., 2019; Bashivan et al., 2019; Walker et al., 2019; Gu et al., 2022; Cowley et al., 2023; Willeke et al., 2023; Pierzchlewicz et al., 2024; Wang and Ponce, 2024). Unlike DNN units where the preferred features are enough to characterize a unit’s tuning (Fig. 3c, 'pref-only' above 'random', <4k images), a V4 model neuron required both preferred and anti-preferred images to outperform randomly-chosen images (Fig. 3b). This suggests that one cannot simply characterize a V4 neuron’s tuning based on its preferred image (a common practice in the field). We find that the anti-preferred feature of a neuron is largely independent of its preferred feature (Fig. 4 and new analyses). We now clarify these points in the manuscript.
> >
> > > Could you elaborate on the possible biological mechanism that gives rise to these two-tailed response distributions? For example, do you suspect inhibitory circuitry or normalization processes in V4 as the source of the anti-preferred suppression?
> >
> > V4 neurons almost always have a large initial transient in response to an image (Fig. 2c, top panel, ‘preferred’). We suspect this transient, which appears independent of stimuli (e.g., averaging over responses to all stimuli reveals a large initial transient), ”pushes” the firing rate into a large dynamic range where stimuli may both excite and suppress that baseline rate. The neural mechanisms behind this transient and its purpose remains largely a mystery in neuroscience (adaptation to repeats of the same stimulus appear to affect it), but our work may shed light on its existence. We now add this point to the Discussion.
> >
> > > You show that combining pre-ReLU features improves prediction. Have you tried using activation functions other than ReLU (e.g., tanh or leaky-ReLU) to see whether two-tailed behavior naturally emerges in artificial networks?
> >
> > We did not try other activation functions, as the point we try to make in the Fig.2a analysis is not about activation function specifically but rather how the different ways of splicing the two-tails of responses (which is the case for pre-ReLU) can affect the predictions. Thus, our goal in Fig.2a was to highlight that current DNNs do not capture the biological response representations found in real neurons.  And although our analysis in Fig.2a is not the conclusive proof of the existence of anti-preferred images, we believe it is a nice corollary of anti-preferred responses.
> >
> > > The ImageBeagle tool seems promising. Can you clarify how well it generalizes across different neurons or datasets? For instance, would it work for V1 or IT recordings, or for neurons modeled with non-DNN representations?
> >
> > We assume that the ResNet50 distance estimation is enough for almost all of the visual tasks needed by visual neuroscientists. The ResNet50 embeddings are enough to find human perceptually similar nearest neighbors; we include example nearest neighbors for randomly-chosen images in the Appendix, see Supplementary Figure 4.  We will also include the results for V1 and IT neurons to supplement our V4 results and include them in our camera-ready version. Based on preliminary results, ImageBeagle’s performance remains the same.

---

> > > ### Author Response · Authors · 2025-11-21
> > > **Questions [Cont.]**
> > >
> > > > In the human experiments, participants performed better when shown both preferred and anti-preferred images. Do you think this reflects an actual perceptual analogy to how V4 represents visual information, or is it more an artifact of the task design?
> > >
> > > This cannot be due to task design, as our control of the DNN units (Fig. 4c) indicates that preferred images only are enough to achieve the best prediction by human participants in this setting; each participant performed both tasks (without knowing which one was which). Thus, participants truly relied on anti-preferred images to improve their decisions. The human participants are another way to validate the importance of anti-preferred images, complementary to our data pruning results (Fig. 3b). That both human and model perform better at predicting a neuron’s tuning by including anti-preferred images suggests anti-preferred images are an important property of the V4 neuron’s tuning. Human participants likely relied on high-level strategies to infer a neuron’s tuning (e.g., “it likes red but doesn’t like horizontal edges”) versus the precise tuning computations. It is an interesting open question how much training a human needs to perform the task close to perfect, suggesting the human has access to similar V4 neurons as the one in the task.

---

> > > > ### Comment · Reviewer_Bncs · 2025-11-26
> > > >
> > > > Thank you for your  responses. While several points are clarified, I still find that some of the core questions remain only partially addressed.
> > > >
> > > > 1. Generality beyond ResNet-50
> > > > You show skewness for other architectures, but this does not fully address whether your main result, especially the predictive role of anti-preferred stimuli, the pre-ReLU mapping argument, and the ImageBeagle behavior, hold across models. Reporting only skewness values is not enough to demonstrate that the central analyses generalize. It would be helpful to see at least some replication of the tuning or data-pruning analyses for another non–ResNet50 model, even if on a smaller scale.
> > > >
> > > > 2. Independent informational value of anti-preferred stimuli
> > > > The response still does not fully resolve the conceptual issue:
> > > >
> > > > * Anti-preferred images alone predict no better than random.
> > > >
> > > > * Yet the paper claims they are a key part of encoding.
> > > >
> > > > Your rebuttal says they are “complementary,” but this still leaves unclear what specific information they add that preferred images alone do not provide. If anti-preferred stimuli are as important as the paper claims, it would be valuable to clarify what actual encoding dimension or structure they capture that is not accessible from the preferred tail.
> > > >
> > > > Human psychophysics interpretation
> > > > Your answer argues that the effect is not a task artifact because DNN units do not show the same behavior. However, this comparison does not fully rule out alternative explanations—e.g., differences in structure or variability of the stimuli shown for V4 vs. DNN units. The behavioral result remains interesting, but I still believe more discussion is needed to justify its interpretation as evidence for biological relevance rather than task structure.
> > > >
> > > > Thank you again.

---

> > > > > ### Author Response · Authors · 2025-12-01
> > > > > **Response to Reviewer Bncs**
> > > > >
> > > > > > Generality beyond ResNet-50 You show skewness for other architectures, but this does not fully address whether your main result, especially the predictive role of anti-preferred stimuli, the pre-ReLU mapping argument, and the ImageBeagle behavior, hold across models. Reporting only skewness values is not enough to demonstrate that the central analyses generalize. It would be helpful to see at least some replication of the tuning or data-pruning analyses for another non–ResNet50 model, even if on a smaller scale.
> > > > >
> > > > > We agree that the skewness alone is not an indicator of the role of anti-preferred images. The skewness analysis is rather a demonstration of the initial divergence between the biological responses and DNN activations. As we mentioned in our earlier rebuttal to the Reviewer, to make this generalizability more concrete, we will include the same analyses for another predictive model, AlexNet, in the final paper.  We will also include the results for V1 and IT neurons to supplement our V4 results and include them in our camera-ready version. Based on preliminary results, ImageBeagle’s performance remains the same.
> > > > >
> > > > > > Independent informational value of anti-preferred stimuli The response still does not fully resolve the conceptual issue:
> > > > > Anti-preferred images alone predict no better than random. Yet the paper claims they are a key part of encoding.
> > > > > Your rebuttal says they are “complementary,” but this still leaves unclear what specific information they add that preferred images alone do not provide. If anti-preferred stimuli are as important as the paper claims, it would be valuable to clarify what actual encoding dimension or structure they capture that is not accessible from the preferred tail.
> > > > >
> > > > > The existence of anti-preferred images itself fundamentally changes how we view the neural computations of V4 neurons as anti-preferred and preferred images are more informative of a neuron’s tuning than solely its preferred images. That “anti-preferred images alone predict no better than random” is not unexpected for our analysis in Figure 3, preferred images also predict worse than random. This indicates that these sets of images are only a component of a neuron’s tuning, including both preferred and anti-preferred components is enough to beat random-selection.  We want to reiterate our further point that the goal of Figure 3 is not to show that anti-preferred images alone are better predictors but rather to show that a neuron’s tuning is more than its preferred images. While we speculate (and now add new analyses) why anti-preferred features may improve the coding capacity of V4 neurons, this is largely out of the scope of our work; instead, we believe our work will spark new experiments and computational studies to understand why visual cortical neurons have two preferences when a single preference appears more than enough in typical DNN architectures. We note that the discovery of grid cells in the hippocampus was first a mystery that sparked decades of effort to understand how these cells’ encoding contributes to spatial navigation. Regardless of the role of anti-preferred features in feature coding in the visual cortex, that V4 neurons have anti-preferred features as an important property of their tuning advances our understanding of feature processing in the brain and may allow us to rethink feature processing in DNNs.
> > > > >
> > > > > > Human psychophysics interpretation Your answer argues that the effect is not a task artifact because DNN units do not show the same behavior. However, this comparison does not fully rule out alternative explanations—e.g., differences in structure or variability of the stimuli shown for V4 vs. DNN units. The behavioral result remains interesting, but I still believe more discussion is needed to justify its interpretation as evidence for biological relevance rather than task structure.
> > > > >
> > > > > We pass the same set of images to our V4 model neurons and DNN units in our analysis in Figure 4. What is different between them is how these models internally represent these images as preferred and anti-preferred. Thus, we do not believe that the differences in structure, or variability of stimuli could lead to the difference in results, but instead how the models themselves represent these images. The key point remains is that human performance improvement when both anti-preferred and preferred images are shown cannot be attributed to task structure or stimulus variability, because DNNs given the same images do not show this pattern.

---

### Author Response · Authors · 2025-11-21
**Global Response**

We thank the reviewers for their time and feedback.

Overall, the reviewers found our work to be an important contribution to neuroAI, aligning deep neural networks with biological neurons. The existence of anti-preferred images opens up new avenues of research into more biologically founded activation functions. Moreover, the reviewers found the extensive suite of methods we used to propose the existence of anti-preferred images (experimental validation, human psychophysics, data pruning) robust,  well presented, and addressing an important question at the intersection of biological and artificial vision. The existence of anti-preferred images challenges our assumptions about neural tuning, especially about feature selectivity in higher-order visual cortex. We add two major new analyses based on Reviewer comments, which we describe in the next paragraph. We also respond to each of the  Reviewer’s comments point by point.


We will add two new analyses to our camera-ready paper. These two analyses will further support our claims that “preferred and anti-preferred images of a neuron have little to no relationship” and consequently “anti-preferred images double the capacity of V4 population for feature selectivity”. The first analysis is a classification task in which the classifier must identify a neuron’s nearest neighbor based on their preferred and anti-preferred images. We find this kNN classifier performs close to chance levels, confirming there is little to no relationship between preferred and anti-preferred features. Why, then, do V4 neurons have anti-preferred features? In our next analysis, we decode the population of V4 neuron models to perform an object recognition task (Caltech 101 image dataset). We find that to achieve the same accuracy, a population of V4 models with one-tailed responses requires double the number of neurons versus the population with original two-tailed responses. This suggests that anti-preferred features provide another entire set of features for downstream neurons to use…effectively doubling the capacity for feature selectivity for the same number of neurons.


We hope to have a revised version uploaded before the December 3rd deadline, however if the full analyses are not complete by then, we will include both of these analyses in the camera-ready version. We are happy to provide further details of these (and our other) analyses to the Reviewers, as requested.

---

### Author Response · Authors · 2025-12-03
**Revised manuscript**

We thank the reviewers and ACs for their time and feedback.  We uploaded a revised manuscript with our new analyses (now Figure 5) where we further investigate the relationship between preferred and anti-preferred images to support our claims that preferred and anti-preferred images of a neuron have little to no relationship (Fig. 5a-d). We then show, via a simple object classification task, that by having preferred and anti-preferred images that have little relationship, the V4 population doubles its capacity for feature selectivity (Fig. 5e-f). We would like to note that this is not the final manuscript, as we will revise the manuscript further to incorporate all the changes discussed in the rebuttal. However, we were excited to share these new analyses that further highlight the role of anti-preferred images, so we prioritized our efforts to get this figure up before the December 3rd deadline.

---

### Meta-Review · Area_Chair_GY4o · 2026-01-12

**Summary:**

This paper investigates “anti-preferred” stimuli in the macaque visual system—images that actively suppress neural firing rather than merely failing to excite it. By integrating electrophysiology, computational modeling, and human psychophysics, the authors demonstrate that V4 neurons exhibit two-tailed response distributions. This fundamentally contrasts with the one-tailed, ReLU-like activation patterns prevalent in standard deep neural networks (DNNs).

Scientific Claims and Findings
The authors establish that anti-preferred images are highly structured and carry distinct information about a neuron’s tuning. They show that while standard DNN units act as one-tailed detectors, biological neurons utilize both excitation and suppression. A key finding is that preferred and anti-preferred features are largely independent, effectively doubling the feature selectivity capacity of the neural population.

Strengths and Weaknesses
Reviewers Bncs and c1B5 praised the robust experimental design and the integration of diverse methodologies. The introduction of ImageBeagle, a tool for efficient large-scale natural image search, was highlighted as a significant practical contribution to the NeuroAI community. However, Reviewer iGeu initially questioned whether the improved predictive performance of the proposed linear-ReLU-linear mapping was simply due to increased model capacity. Furthermore, Reviewer qR7N noted that anti-preferred stimuli alone did not outperform random sampling in specific predictive tasks, suggesting their value is primarily complementary rather than independent.

Justification for Decision
The authors successfully addressed concerns regarding model capacity by providing capacity-matched controls and new decoding analyses. The consensus among the majority of reviewers is that this work identifies a critical biological divergence in modern AI architectures. By challenging the ubiquity of the ReLU-like abstraction, the paper offers timely insights into building more biologically realistic and efficient vision models. The alignment of human psychophysics with model predictions further strengthens the claim that two-tailed tuning is a central property of biological visual representation.

**Reviewer Concerns:**

The authors did a remarkable job in addressing reviewer's questions.

**Reviewer Scores:**

Given the thoroughness of the author's responses, I am confident that the scores would have been better upon participation in a full discussion.

---

### Decision · Program_Chairs · 2026-01-26

Accept (Poster)